# Wafer-scale integration of two-dimensional perovskite oxides towards motion recognition

Ming Deng[1], Ziqing Li [2] ✉, Shiyuan Liu[3], Xiaosheng Fang [1,2] ✉ & Limin Wu [1,4] ✉

Two-dimensional semiconductors have shown great potential for the development of advanced intelligent optoelectronic systems. Among them, two-dimensional perovskite oxides with compelling optoelectronic performance have been thriving in high-performance photodetection. However, harsh synthesis and defect chemistry severely limit their overall performance and further large-scale heterogeneous integration. Here, we report the wafer-scale integration of highly oriented nanosheets by introducing a charge-assisted oriented assembly film-formation process and confirm its universality and scalability. The shallow-trap dominance induced by structural optimization endows the device with a distinguished performance balance, including high photosensitivity close to that of single nanosheet units and fast response speed. An integrated ultra-flexible 256-pixel device demonstrates the versatility of material-to-substrate integration and conformal imaging functionality. Moreover, the device achieves efficient recognition of multidirectional motion trajectories with an accuracy of over 99.8%. Our work provides prescient insights into the large-area fabrication and utilization of 2D perovskite oxides in advanced optoelectronics.

Two-dimensional (2D) semiconductor materials have been thriving in the most advanced optoelectronic devices such as machine vision[1–3], bio-inspired neuromorphic visual systems[4–7], and in-sensor computing architectures[8,9] due to their tunable properties in optics and optoelectronics and extensive applicability for heterogeneous integration. Both experimental and theoretical investigations on 2D semiconductors have enabled critical fundamental understanding of this material system. Among them, 2D perovskite oxides (e.g., $Ca_2Nb_3O_{10}$, $Sr_2Nb_3O_{10}$, and so on) with unique physicochemical properties and compelling optoelectronic performance have been emerging as up-to-date frontiers in ultraviolet (UV) photodetection[10–15]. While possessing the valuable features of 2D perovskite, such as tunable bandgap,

superior photoconductive gain, and efficient carrier extraction under atomic-scale thickness[12,13,16], 2D perovskite oxides overcome the disadvantage of instability and toxicity of the mainstream metal halide perovskites[17]. More importantly, they compensate for the scarcity of common 2D semiconductors in the field of high-performance UV detection, since most eye-catching 2D semiconductors with narrow bandgaps are oriented toward visible and infrared light (Supplementary Table 1).

The current prosperity of stellar 2D semiconductors (e.g., $MoS_2$[8,18], $WS_2$[19], and $WSe_2$[20]) in optoelectronics closely connects with their feasibility of large-area fabrication and amenability to heterogeneous integration with photonic circuits and microelectronics[21]. In

[1]Department of Materials Science and State Key Laboratory of Molecular Engineering of Polymers, Fudan University, Shanghai, P. R. China. [2]Shanghai Frontiers Science Research Base of Intelligent Optoelectronics and Perception, Institute of Optoelectronics, State Key Laboratory of Photovoltaic Science and Technology, Fudan University, Shanghai, P. R. China. [3]Optical Fiber Research Center, Department of Materials Science, Fudan University, Shanghai, P. R. China. [4]College of Chemistry and Chemical Engineering, Inner Mongolia University, Hohhot, P. R. China. ✉e-mail: lzq@fudan.edu.cn; xshfang@fudan.edu.cn; lmw@fudan.edu.cn

contrast, the harsh synthesis and defect chemistry of 2D perovskite oxides have been two of the main obstacles to their progression[10,22]. Severe synthesis conditions make large-area preparation with controllable thickness unattainable, and the persistent photoconductivity effect induced by the abundant traps leads to slow response speeds, making it difficult to strike a balance between high responsivity and fast response speed. Existing studies have shown the marvelous capability of 2D perovskite oxides in photodetection, which have triggered extensive attention[14,15,23–26]. However, more in-depth consideration of the above two issues could be beneficial. Most current research focuses on the performance enhancement of individual nanosheets or devices based on the top-down strategy, which might not fully clarify the direction for future investigations and applications of such valuable materials[13]. Therefore, while there is an urgent need to find strategies to achieve a performance balance, large-area or wafer-scale integration is expected to accelerate the research and commercialization of 2D wide-bandgap semiconductors for advanced optoelectronics. In addition, the development of a universal approach to expand the future integration of 2D perovskite oxides with CMOS, TFT, and flexible electronics is also critical to their research and development.

In this work, we introduce a charge-assisted oriented assembly film-formation process to achieve large-area ordered spatial orientation of nanosheet functional units. Through this process, wafer-scale integration with controllable thicknesses of 2D perovskite oxide semiconductors toward optoelectronics is realized. The highly ordered orientation of the nanosheets and the resultant shallow-trap dominance entitle the device with significantly faster response speed compared with those of reported 2D perovskite oxide-based photodetectors and performance superior to that of state-of-the-art perovskite UV devices. Subsequent large-area heterogeneous integration of an ultra-flexible 256-pixel array demonstrates the integration versatility of the strategy and its application potential in advanced electronics. Performance optimization enables the device to output high-

quality single-frame images, solving the common ghosting issue. In conjunction with the designed program processing and a trained convolutional neural network (CNN) model, efficient motion recognition is realized using the obtained spatiotemporal images. This work provides a universal strategy for the effective large-scale integration of 2D perovskite oxide semiconductors and sheds light on the future application and industrialization of this class of materials for advanced microelectronics.

## Results

### Charge-assisted oriented assembly film-formation process

Ca₂Nb₃O₁₀ (CNO) nanosheets are obtained through a traditional top-down process: high-temperature solid-phase calcination, ion exchange, and liquid-phase exfoliation (Supplementary Fig. 1). Given the special intrinsic negative charge of CNO (Supplementary Fig. 2), to achieve the large-area homogeneous preparation, we introduce a charge-assisted oriented assembly film-formation (COAF) strategy. In contrast to the random settling and haphazard stacking of nanosheets in the dipped CNO (D-CNO) film[14,23,24] (Supplementary Fig. 3), the COAF process provides the possibility to produce large-area ordered thin film. Our assembly process adopts green and pollution-free water-ethanol cosolvents instead of the organic solvents (DMSO, DMF, etc.) used in most studies for dissolving and dispersing perovskite-type 2D materials. The cosolvent systems are introduced to tailor the dispersion of nanosheets and the volatilization rate of solvents. The solvent mixture of deionized water and ethanol not only provides reduced surface energy for better dispersion of solutes and enhanced wettability of the solution on substrates (Supplementary Fig. 4), but also enables the nanosheets to fast stack layer-by-layer with the rapid volatilization of solvent to avoid the percolation issue. As shown in Fig. 1a, in this process, the sprayed large number of micro-droplets first ensure the large-area in-plane spreading on the substrate. The repulsive forces induced by the intrinsic negative charges between the nanosheets in micro-droplets prevent aggregation and facilitate their

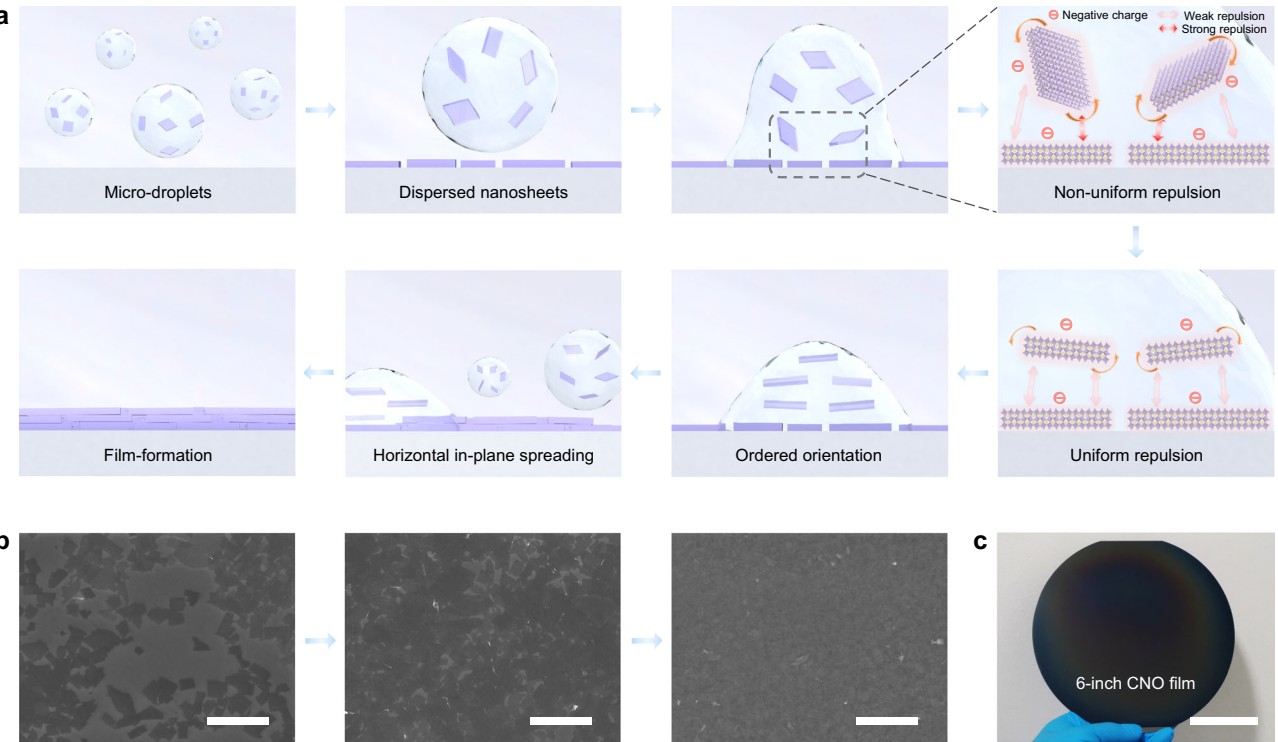

**Fig. 1 | Charge-assisted oriented assembly film-formation process. a** Schematic of the charge-assisted oriented assembly film-formation (COAF) process of the S-CNO film. Specific interpretations are presented in Supplementary Note I. **b** Surface morphology SEM images of the S-CNO film during the COAF process. Scale bar: 1 μm. **c** Photograph of the integrated large-area S-CNO film on a 6-inch n-type Si wafer. Scale bar: 5 cm.

dispersion. Then, when the droplet contacts the substrate, non-uniform repulsive forces between the nanosheets in the droplet and those on the substrate induce the former to rotate to be approximately parallel to the latter in order to balance the repulsive forces, resulting in a highly ordered orientation. As the solvent evaporates, the nanosheets in each micro-droplet settle and stack layer by layer, accompanied by the filling of voids and gaps between the larger nanosheets by the tinier ones (Supplementary Fig. 5). But given the high shape irregularity of the nanosheets, the formed films may still not be totally void-free. After the complete evaporation of the solvent, tight structures with a high degree of orientation are formed. Meanwhile, since the large number of micro-droplets cover the substrate surface uniformly over a large area during the spraying process, the nanosheets within each droplet will connect with those adjacent ones to form an integral whole. Ultimately, the ordered stack and the connections covering the whole substrate of the nanosheets simultaneously complete the construction of the highly oriented film. More details are discussed in Supplementary Note I. And Fig. 1b shows the corresponding surface morphology scanning electron microscopy (SEM) images of the film during the COAF process.

In addition to thickness control and ordered orientation, this strategy also allows for heterogeneous integration of CNO films at the wafer scale (on a 6-inch n-type Si wafer), as shown in Fig. 1c. This solves the problem of large-area preparation of 2D perovskite oxide films and holds promise for future large-scale integration of such materials with microelectronics and circuits. Another prerequisite for integration applications is uniformity, as film morphology and surface defects significantly affect light absorption and carrier transmission processes. Optical microscope photographs and SEM characterization of surface morphology in Supplementary Fig. 6 and Supplementary Fig. 7 show that compared with the rough and uneven surface with obvious protrusions and steps of the D-CNO film, sprayed CNO (S-CNO) films are significantly flatter and more homogeneous. In particular, the detailed SEM characterizations also illustrate the formation process of dense films through the ordered stacking of nanosheets, corroborating the proposed COAF process. Quantitative roughness characterized by atomic force microscope (AFM) also reveals that the surfaces of the S-CNO films are significantly smoother (Supplementary Fig. 8), indicating fewer defects and more favorable carrier transport. Incidentally, these images still show some relatively large particles, which are believed to be thicker nanosheets and incompletely exfoliated $HCa_2Nb_3O_{10}$ (HCNO), and better optimization of film structure is expected to be realized by optimizing the synthesis or adopting the centrifugation method to obtain nanosheets with a uniform distribution of thickness and size.

Subsequently, comprehensive characterizations of the nanosheet films are performed. With transmission electron microscopy (TEM) and X-ray photoelectron spectroscopy (XPS) tests, we demonstrate the purity and homo-disperse of the elements (Supplementary Fig. 9 and Supplementary Fig. 10). Intriguingly, while the X-ray diffraction (XRD) pattern of the D-CNO film shows various diffraction peaks arising from different crystal planes, the S-CNO film magically exhibits only equivalent crystal planes of (001), (006), (007), and (0010), illustrating that the nanosheets are all aligned alone [001], as shown in Fig. 2a. Accordingly, we plot the micro-schematic of the cross-section of the nanosheet films, as shown in Fig. 2b. Efficient carrier transport between nanosheets requires decent relative alignments between adjacent ordered crystalline regions[27]. The nanosheets with preferred orientation in the S-CNO film are more conducive to the transportation and drift of carriers.

The excitation-emission matrix (EEM) spectrum plotted in Supplementary Fig. 11 illustrates the photoresponse of CNO in the UV region, which is consistent with the absorption spectra and the wide bandgap of 3.61 eV in Supplementary Fig. 12. The films exhibit evenly increased UV absorption and decreased transmittance as they become thicker. It is considered that the densification and homogeneity of the

film, i.e., the compactness and orderliness of the stacked nanosheets in this research, have a major impact on the carrier diffusion and recombination in CNO films. The steady-state and time-resolved photoluminescence (PL) characterizations in Fig. 2c expound the dynamics of photogenerated carriers in films. The quenched PL intensity of the S-CNO film indicates fewer recombination sites, as the diminished surface and bulk defects in the dense film result in a lower carrier recombination rate, which is conducive to efficient photodetection[22]. Note that the non-unimodal PL spectra originate from multiple signals generated by recombination, exciton binding, and $NbO_6$ groups at different sites[24]. Meanwhile, as the PL lifetime is predominantly determined by defect-induced non-radiative recombination, a longer lifetime corresponds to a decreased number of non-radiative recombination centers[28], as shown in Supplementary Fig. 13. The regular stacking of nanosheets results in weaker impediments of the electric field to the horizontal motion of the carriers and lower probabilities of recombination at defects and interfaces, ultimately leading to a longer carrier lifetime.

In addition, we test the grazing incidence wide-angle X-ray scattering (GIWAXS) patterns to further verify the above analyzes. As depicted in Fig. 2d–f, the patterns of the S-CNO film exhibit concentratively punctate and elliptical signals. The shrinkage of the signal region under certain grazing incidence angles confirms the highly ordered orientation of the nanosheets in Fig. 2b. The displayed concentrated signals with similar intensities appearing in a single direction correspond to the equivalent peaks in the XRD results. In contrast, in Fig. 2g–i, the arc-shaped signals representing the smaller interplanar spacings imply a disordered out-of-plane orientation but in-plane isotropy in a portion of the film. And the rings of uneven intensity indicate the inhomogeneous distribution of the film in such a disordered state. Moreover, by increasing the grazing incidence angle from 0.1° to 0.2° and 0.4°, we probe the average orientation condition and depth profile deeper into the films. It is found that for D-CNO film, the GIWAXS signal region expands with increasing angle, whereas that of the S-CNO film remains constant, which further demonstrates the consistent and regular orientation of the S-CNO film at different depths. Supplementary Fig. 14 also confirms that the increase in film thickness cannot interfere with the alignment of the nanosheets. In conclusion, the GIWAXS results are consistent with the XRD results and simultaneously validate the proposed COAF process.

Moreover, we prepare a large-area nanosheet film of another representative material, $Sr_2Nb_3O_{10}$, to confirm the universality of the fabrication strategy for a large class of 2D perovskite oxides (Supplementary Fig. 15). XRD patterns for different positions of the wafer-scale $Sr_2Nb_3O_{10}$ film also exhibit all equivalent planes aligned alone [001].

## Performance balance achieved by shallow-trap dominance

Structure optimization in the film always leads to performance improvement in the device, as a homogeneous morphology and ordered structure favor the transport of charge carriers[29,30]. The inhomogeneity of the traditional D-CNO film at both micro- and macro-scale makes the fabricated devices exhibit great variations in performance (Supplementary Fig. 16), making them highly uncontrollable and unavailable. For the S-CNO devices schematically shown in Fig. 3a, the optoelectronic performance (i.e., photocurrent and on-off ratio) first improves and then decays with increasing thickness, as shown in Fig. 3b and Supplementary Fig. 17. This is mainly related to the film thickness, light absorption process, and defects. A more specific interpretation is provided in Supplementary Note II. With a channel area of 50,000 $\mu m^2$, as shown in Fig. 3c, the optimal S-CNO 10 mL photodetector achieves a prime responsivity and a remarkable detectivity of 11.9 A $W^{-1}$ and $3.71 \times 10^{14}$ Jones under 280 nm UV illumination and 5 V bias, outperforming the state-of-the-art perovskite-based UV photodetectors (specific comparisons are shown in Supplementary Table 2). The topmost reported UV rejection

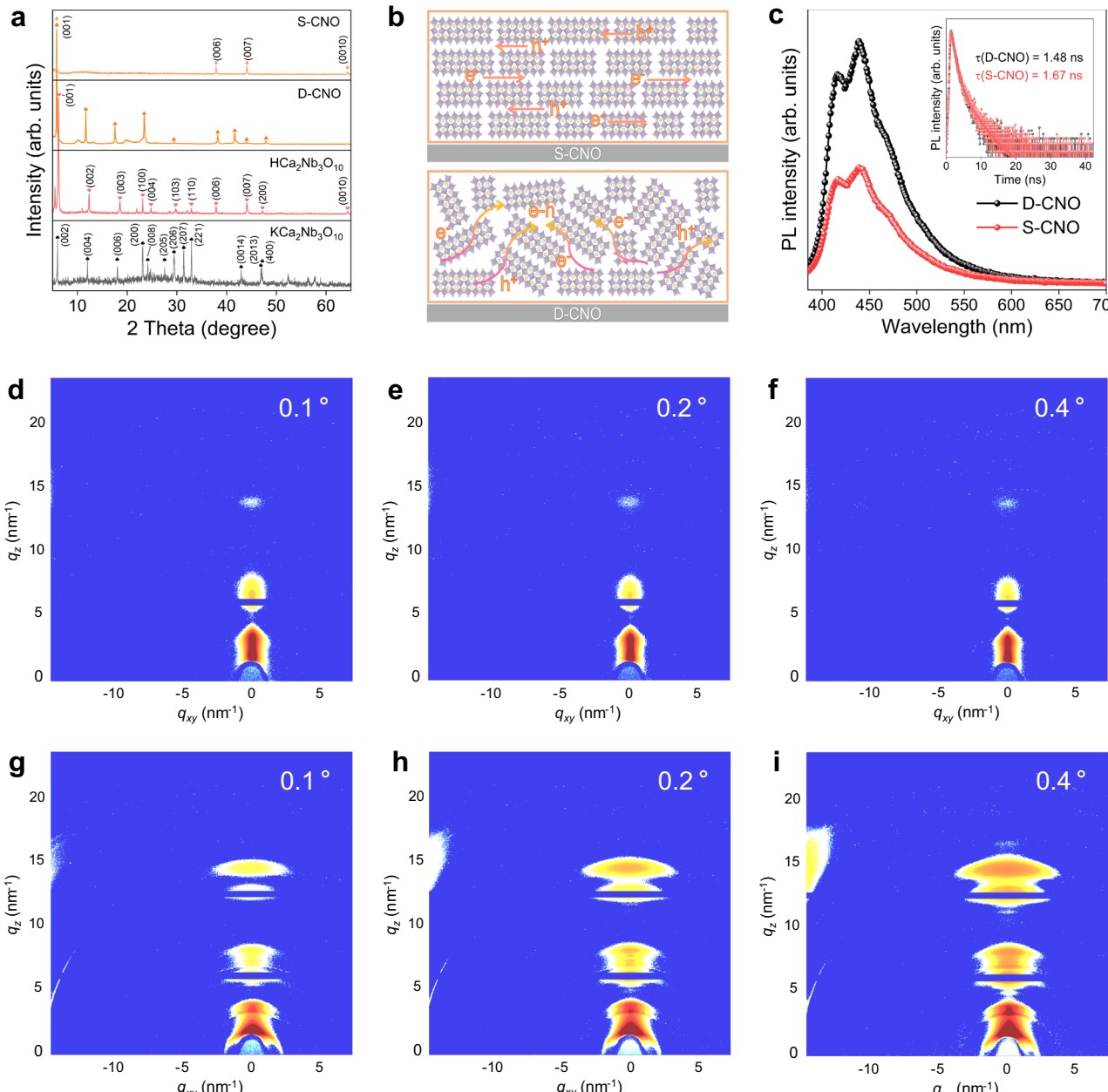

**Fig. 2 | Characterization of D-CNO and S-CNO films. a** XRD patterns of KCNO, HCNO precursors, and D-CNO, S-CNO films. **b** Schematic of the stacking conditions of nanosheets in the formed S-CNO and D-CNO films. **c** Steady-state PL emission spectrum of D-CNO and S-CNO films at room temperature with an excitation wavelength of 365 nm. Inset is the TRPL spectrums of D-CNO and S-CNO films monitored at λ = 440 nm. **d**–**f** GIWAXS patterns of S-CNO films with incident angles of 0.1° (**d**), 0.2° (**e**), 0.4° (**f**). **g**–**i** GIWAXS patterns of D-CNO films with incident angles of 0.1° (**g**), 0.2° (**h**), 0.4° (**i**). Note that all the GIWAXS patterns use the same color bar as shown in Supplementary Fig. 14.

ratio as far as we know of 185,560 displays the ultrahigh selectivity of the CNO device to UV light, which corresponds to the EEM and UV-vis absorption spectra and is of great significance for specific applications[17]. All calculations in this section for performance evaluation refer to Supplementary Note III.

In the field of photodetection, it always remains challenging to balance high photosensitivity with fast speed. This contradiction is more common in 2D perovskite oxides due to their complex defect structures, such as vacancies, antiphase domain boundaries, and local structural collapse[10,13]. While the photogating effect caused by the large surface-volume ratio and numerous defects endows them with high optical gain and photosensitivity, it also introduces ultraslow photoresponse[31,32]. In the traditionally formed CNO nanosheet films, the massive gaps and defects at the interfaces of nanosheets severely

hinder carrier transport, resulting in a significantly slower response speed[23,24]. However, in our work, because of the fast carrier transport process arising from the ordered orientation of nanosheets, the S-CNO device realizes substantially improved response speed while maintaining ultrahigh photosensitivity, as shown in Fig. 3d. More systematic tests of the response time of various S-CNO devices better illustrate their overall significant improvement compared with the traditional device (Supplementary Fig. 18 and Fig. 3e). Typically, a response speed of $t_r = (42.4 \pm 14.8)$ μs and $t_d = (1.77 \pm 0.40)$ ms of the S-CNO 10 mL device is 10 and 20 times faster than that of the D-CNO device, respectively, which is also significantly faster than those of reported 2D perovskite oxide-based photodetectors.

The significantly increased speed can be attributed to the optimized film structure, which is consistent with the morphological

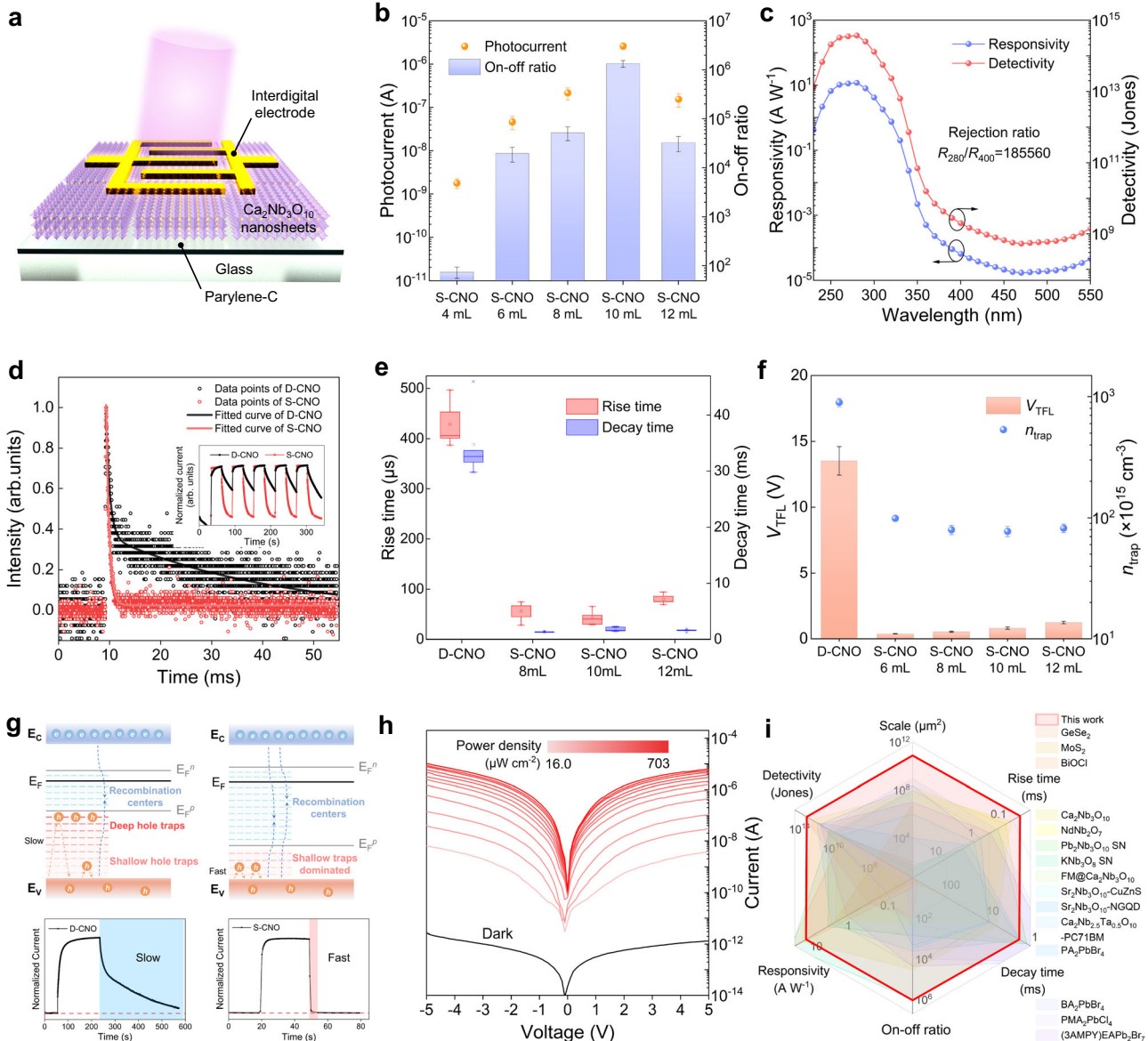

**Fig. 3 | Optoelectronic characteristics of the S-CNO film photodetector.**
**a** Schematic illustration of the S-CNO photodetector architecture. **b** Photocurrents and on-off ratios of S-CNO devices measured under 5 V bias and 280 nm UV illumination (703 μW cm$^{-2}$). The error bars represent the standard deviation and are calculated based on the variation in the photocurrent and the on-off ratio.
**c** Spectral responsivity and detectivity of the optimal S-CNO 10 mL photodetector illuminated by monochromatic light with wavelength varying from 240 to 550 nm at 5 V bias. **d** Comparison of the photoresponse speed of transient and $I$-$t$ curves (inset) of D-CNO and S-CNO devices. **e** Statistics of the output photoresponse time for D-CNO and S-CNO devices. The error bars represent the standard deviation and are calculated based on the variation in the response time. **f** Statistics of the $V_{TFL}$ and $n_{trap}$ calculated from every three hole-only D-CNO and S-CNO devices. The

error bars represent the standard deviation and are calculated based on the variation in $V_{TFL}$ and $n_{trap}$. **g** Schematic diagrams of carrier recombination and trapping kinetics in D-CNO and S-CNO films. The schematic illustrations show the band structure containing trap and recombination centers, and the line graphs show the practical photo-switching response. **h** Photoresponse of the S-CNO device in dark and under 280 nm illumination with light intensities varying from 16 to 703 μW cm$^{-2}$. **i** Performance comparison of representative 2D perovskite oxide-, 2D halide perovskite- and other 2D material-based film UV photodetectors, in which the data are concluded from Supplementary Table 2. Note that the performance data in Fig. 3 and corresponding context are measured on D-CNO and S-CNO devices with channel areas of 50,000 μm$^2$.

characterization in the previous section. To provide a better illustration from a mechanistic point of view, we quantify the trap density ($n_{trap}$) by the space-charge-limited current (SCLC) method. As shown in Supplementary Fig. 19, the $I$-$V$ curves of tailor-made hole-only S-CNO and D-CNO devices (ITO-PEDOT-CNO-MoO$_3$-Ag) can be divided into three typical SCLC regions, and the $n_{trap}$ of S-CNO devices that calculated from trap-filled limit voltages ($V_{TFL}$) is significantly lower than that of D-CNO device (Fig. 3f). Figure 3g illustrates a comprehensive interpretation of the accelerated speed. It is well known that the trap state has a profound effect on the response time of semiconductors

(Supplementary Fig. 20)[33–35]. As depicted in detail in Supplementary Fig. 21 and Supplementary Note IV, the localized states introduced by defects, impurity energy levels, and interfacial states have important effects on carrier trapping, de-trapping, and recombination processes, and they serve as recombination or trap states under illumination. In the D-CNO film that contains a large number of defects and traps, the quasi-Fermi level for holes is located near the deep trap states, in which carriers take a much longer time to be trapped or released, thereby endowing the D-CNO film with a slow response time of several hundred seconds[32,34]. The gradual rise of the $I$-$t$ curve and subsequent platform

correspond to the carrier trapping process and the filled-up state of the trap states, respectively. As for the S-CNO film, the layer-by-layer oriented assembly of nanosheets leads to a slump of traps, especially decremental interfacial traps, indicating that there are fewer additional energy levels. This shifts the hole quasi-Fermi level downward and transforms the original deep traps into recombination centers[32]. Owing to the predominance of shallow traps and the incremental recombination centers, the overall trapping and de-trapping time are greatly reduced, resulting in a significantly faster photoresponse speed. Experimental results and interpretations in Supplementary Fig. 13 also support this perspective.

In addition, we investigate the influence of the incident light intensity on photoresponse behavior. The I-V curves under 280 nm UV illumination with different light intensities in Fig. 3h show that the photosensitivity gradually increases as the light intensity increases. Incidentally, we suppose that the non-minimum currents at 0 V bias occurring in the dark and at low light densities may originate from the mismatch-induced built-in electric field at the interfaces of nanosheets, as well as the relatively slower trapping and de-trapping processes of fewer carriers by defects and traps in these cases. The fitted lines in Supplementary Fig. 17e represent the corresponding relationship between the photocurrent and photodensity under certain biases. As the generation efficiency of photoinduced carriers is proportional to the absorbed photon flux, the photocurrent is depicted as a power function of light intensity. The fitted nonunity indexes of the law ($\theta > 1$) are related to the complicated processes of generation, trapping, and de-trapping in the device, implying complex electron trapping and carrier transport in the processed photodetector. Figure 3i demonstrates the performance comparisons of our device with other representative 2D perovskite oxide-, 2D halide perovskite-, and other 2D material-based UV photodetectors.

## Large-area and flexible integration for imaging

The blossoming of advanced 2D semiconductors is closely linked to the feasibility of their large-scale integration, and their combination with ultra-flexible, ultra-light, and biocompatible electronics has broadened their application prospects[18,36,37]. Meanwhile, in contrast to bottom-up synthesis with corresponding growth processes, the integration of 2D semiconductors with substrates and microcircuits in the top-down growth approach remains challenging. To demonstrate the universality of material-to-substrate integration and the potential of the fabricated device toward advanced optoelectronics, we further construct a large-area 16 × 16 device array on an ultra-flexible Parylene-C substrate. Schematic illustrations and photographs of the fabrication process are shown in Supplementary Fig. 22 and Supplementary Fig. 23. The overall structure of the active area is schematically shown in Fig. 4a, and the inset shows a channel area of 4500 µm² for each individual device. And Fig. 4b illustrates the properties of hydrophobicity, flexibility, and ultra-light weight of the device. More vividly, benefiting from ultra-flexibility, the device exhibits conformability, transferability, and wrinkle-ability for various application scenarios (Fig. 4c). Moreover, the homogeneity of the film imparts high consistency in device performance, as the photocurrents are calculated to be $(9.08 \pm 1.60) \times 10^{-9}$ A with a low variation coefficient ($C_V$) of 17.6% under 280 nm illumination and 1 V bias, as shown in Fig. 4d and Supplementary Fig. 24. Note that the relatively low photocurrents stem from the reduction in device channel and applied bias, and this does not mean the significant performance decline compared with rigid device (Supplementary Fig. 24). High contrast and electrical uniformity entitle the device array with excellent imaging capabilities for patterns and letters (Fig. 4e).

Furthermore, we transfer the array to a PET substrate to demonstrate its flexible performance. As shown in Supplementary Fig. 25, individual pixels exhibit stable photocurrents and are not susceptible to performance degradation due to deformation-induced structural variations. Subsequently, the arrays are bent to 90°, and the I-t curves of fifty pixels in the bending state are randomly measured. As shown in Fig. 4f, the response behaviors of these pixels are essentially identical, and their photocurrents remain highly uniform with a $C_V$ of 17.3%. It can be seen that morphology transformations do not interfere with the performance consistency, and consequently the device arrays in the bending state still show decent imaging result (Fig. 4g). Moreover, ultra-flexibility also allows the device array to conform to substrates of arbitrary shapes. By transferring the array onto a glass hemispherical substrate, a UV-sensitive retina-like device is presented (Fig. 4h), which also exhibits clear and stable imaging capability (Fig. 4i).

## High-quality single-frame images for motion recognition

A typical machine vision system mainly consists of light-sensing and current-stimulation function modules[2], and the performance optimization of the light-sensing module is important for multifunction expansion such as motion recognition and prediction, as high photosensitivity and fast response speed of a photosensor are prerequisites for advanced imaging applications. State-of-the-art motion recognition-related devices require specific functions such as multifunction of sensing and memory[7], encoding temporal vision[38], and neuromorphic reinforcement learning[39]. This is undoubtedly an obstacle for numerous conventional image sensors that can only output single frames. Meanwhile, as neural networks such as CNN, ANN, and RNN models play an increasingly important role in the field of artificial vision[1,3–6,40], there are also greater demands for processable high-quality single-frame imaging, which can be utilized for spatiotemporal motion sensing, trajectory recognition, and so on[41]. On the basis of the excellent capability of our fabricated device for single-frame imaging, we propose a model for motion trajectory recognition. As the schematic processing flow in Fig. 5a shows, when the target object moves (taking a falling straw hat as an example), the pixels in the sensor that receive the light signal generate corresponding current changes (simulating visual stimulations), thereby outputting single-frame images with spatial information. Differing from other conventional photosensors, by encoding temporal information into the images by weighting photoresponse currents by time, motion trajectory images with spatiotemporal information can be obtained. Based on the data obtained by the photosensor, we generate a tailor-made dataset consisting of eight directions of motion to train our designed CNN model. Figure 5b shows the structure of the CNN model, and the training and parameter optimization process are described in the "Methods" section.

Figure 5c and Supplementary Fig. 26 show the trajectory imaging results achieved by the control conventional D-CNO and experimental S-CNO photosensors. Note that due to the inhomogeneity of the D-CNO film, an individual D-CNO device is used for single-pixel imaging to simulate the object motion. Due to the ghosting phenomenon (so called visual persistence expounded in Supplementary Fig. 27 and Supplementary Note V) of the control device, the CNN model fails to recognize the trajectory in images obtained by the D-CNO device. The interpretation of the generation of ghosting is displayed in Fig. 5d. While all photosensors follow a process of activation and recovery as they are illuminated and subsequently removed from light, control devices with slow speed take longer to return to their initial states after the light stimulus disappears, thus inducing residual electrical signals, i.e., persistent visual perception. In contrast, the experimental pixels can be activated and reset quickly. As the motion process proceeds, activated pixels directly generate electrical signals, while inactivated pixels get immediately restored after being activated by light. Afterward, the overlap of frames realized by processive perceiving and time-weighting can successfully merge spatial information from each time point and show the spatiotemporal motion trajectory. Figure 5e demonstrates the recognition accuracy of the CNN model for motion trajectories in all eight directions. After 12 training rounds, the

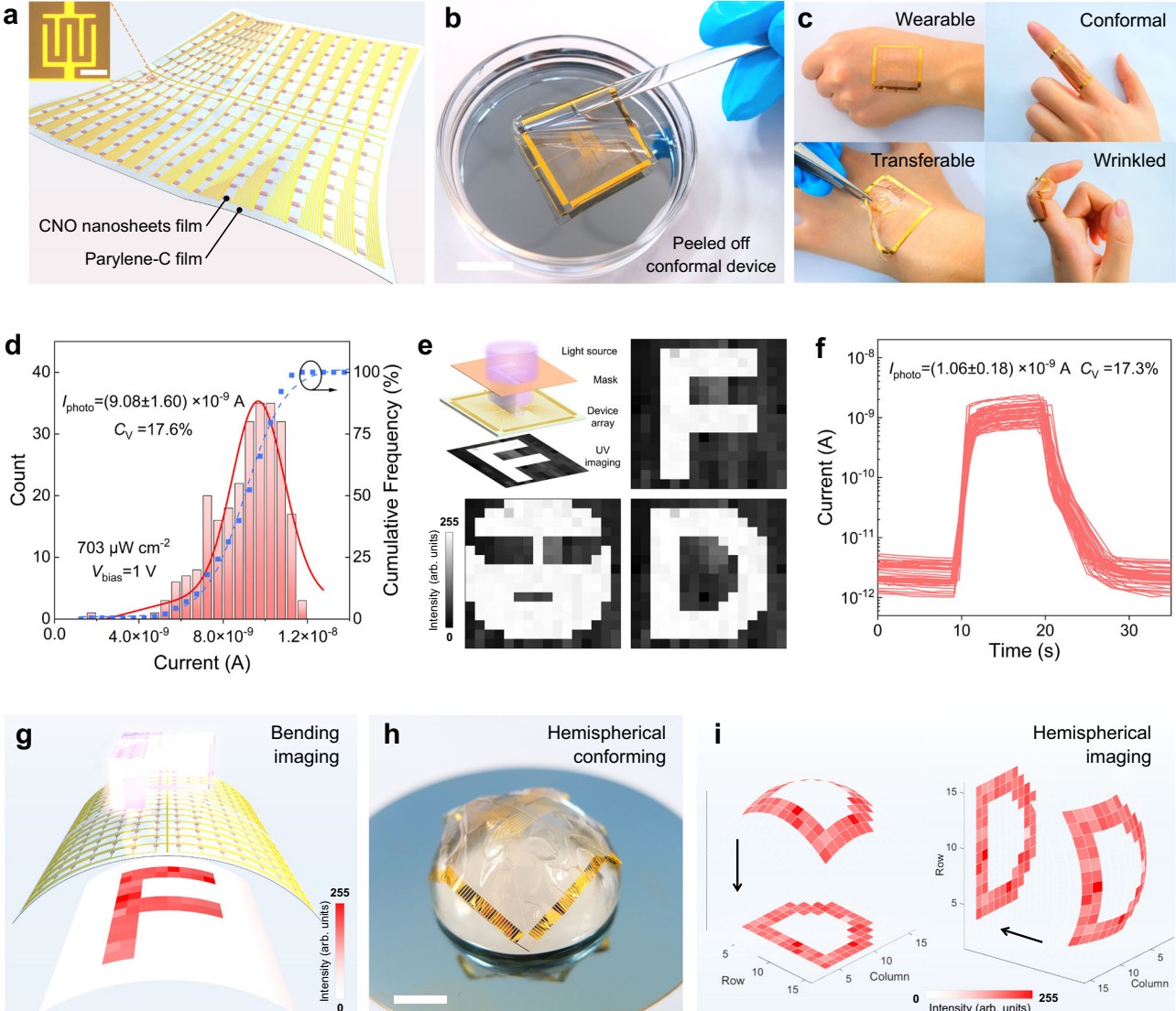

**Fig. 4 | Characterization of the flexibility and imaging capacity of the device array. a** Schematic illustration of the flexible device array. The inset shows an optical microscopy image of an individual S-CNO device with a pair of inter-digital electrodes and a channel area of 4500 μm². Scale bar: 50 μm. **b** Photograph of the peeled-off device array floating on water. Scale bar: 2 cm. **c** Photographs of the device array in wearable, conformal, transferrable, and wrinkled situations. **d** Statistical distribution of the photocurrent of the 256-pixel array under 1 V bias and 280 nm illumination. Scale bar: 1 cm. **e** Schematic illustration of the projection imaging mechanism and corresponding imaging results based on the 16 × 16 device array. **f** I-t characteristics of 50 pixel-units in the 90° bending state. **g** Bending imaging results under 90° bending. **h** Photograph of the flexible array conformed to a hemispherical glass mold. Scale bar: 1 cm. **i** Hemispherical imaging results.

validation accuracy of recognition for the experimental device remains above 99.8%, which is much higher than that of the control device (~60%). Consistent results are also obtained for training accuracy.

## Discussion

In summary, wafer-scale integration of 2D perovskite oxide is realized by a charge-assisted oriented assembly film-formation process. The highly ordered structure orientation and shallow-trap dominance endow the fabricated device with significantly faster photoresponse speed compared with other 2D perovskite oxide-based photo-detectors. The universality and scalability of the strategy for 2D per-ovskite oxides are confirmed by fabricating different materials. Further integration of the ultra-flexible device array demonstrates the versa-tility of material-to-substrate integration and its functionality in com-plicated application scenarios. Moreover, performance optimization has enabled the device to overcome the ghosting issue and output high-quality single frames for spatiotemporal processing, further rea-lizing efficient recognition of motion trajectory in multi-directions.

This work is crucial for the extension of heterogeneous compatibility and advanced applications of 2D perovskite oxide materials.

## Methods

### Top-down synthesis of Ca$_2$Nb$_3$O$_{10}$ nanosheets

The synthesis of Ca$_2$Nb$_3$O$_{10}$ nanosheets consists of the following three main procedures: high-temperature solid-phase calcination, ion-exchange, and TBAOH exfoliation. First, K$_2$CO$_3$ (99.99%), Nb$_2$O$_5$ (99.99%), and CaCO$_3$ (99.99%) with a molar ratio of 1.1: 3: 4 (i.e., a molar ratio of K: Nb: Ca = 1.1: 3: 2) were fully ground for 0.5 h, then heated in a tube furnace for solid-phase calcination to obtain KCa$_2$Nb$_3$O$_{10}$ (KCNO). The reaction temperature was programed to rise to 1,200 °C from room temperature in 2 h, maintain for 12 h, and then naturally cooled to room temperature. Second, after grinding the obtained white pro-duct into powder, 40 mL of HNO$_3$ (5 M) was added for every 0.01 mol of powder (approximately 0.6 g), and the mixture was continuously stirred for four days. Note that HNO$_3$ need to be renewed once a day. The obtained HCa$_2$Nb$_3$O$_{10}$ (HCNO) powder should be washed twice

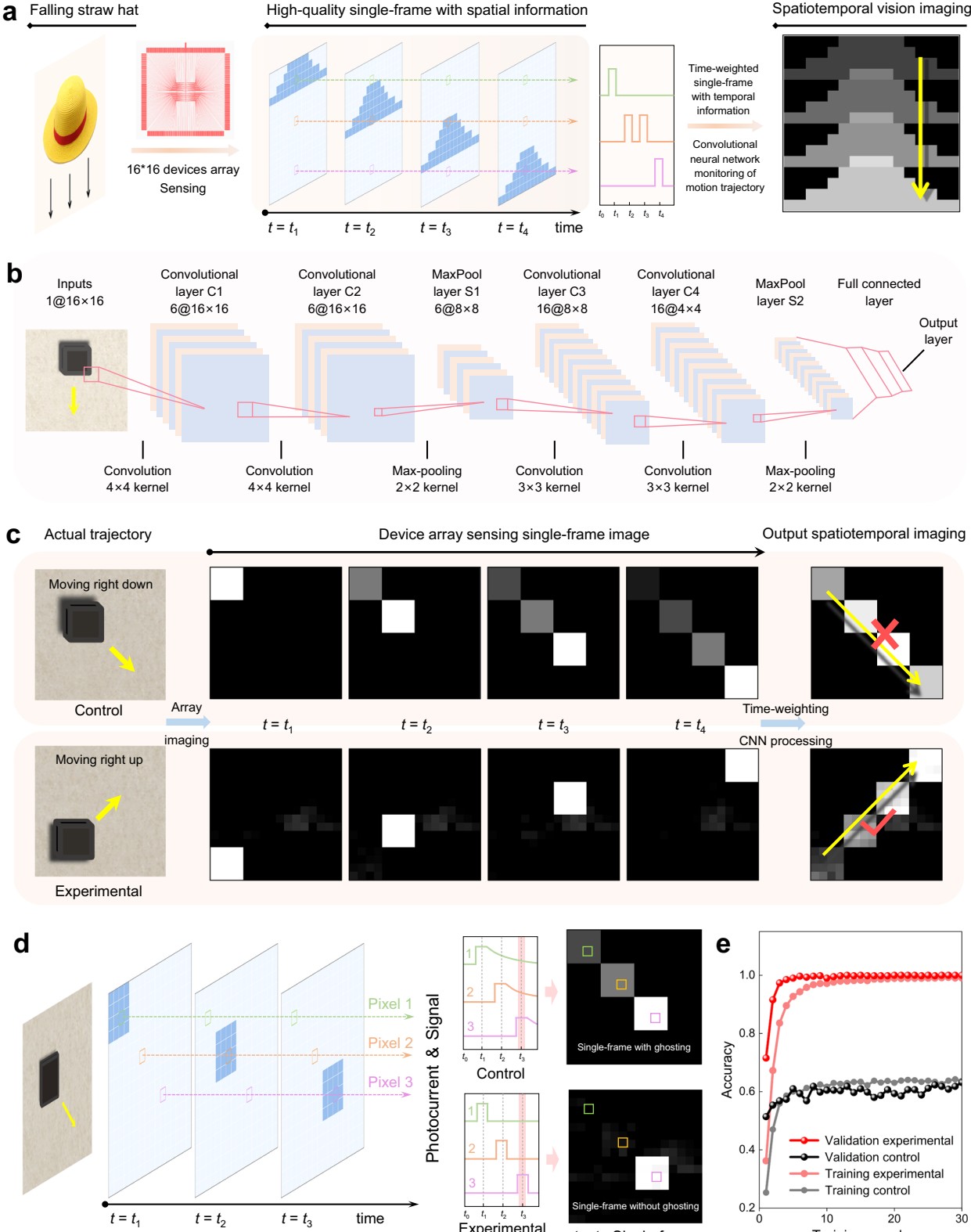

**Fig. 5 | Motion recognition based on device array and CNN processing.**
**a** Schematic illustration of spatiotemporal vision imaging acquisition and the realization of motion recognition. **b** Structure of the designed convolutional neural network model. **c** Simulated current maps and recognition results for left-up to right-down (D-CNO) and left-down to right-up (S-CNO) motion. **d** Interpretation of the single-frame with and without ghosting obtained by the D-CNO and S-CNO devices, respectively. **e** Training and validation accuracy of motion recognition processing achieved by the D-CNO and S-CNO devices.

with deionized water. Third, 0.26 g of TBAOH and 50 mL of deionized water were added for every 0.01 mol of $HCa_2Nb_3O_{10}$ (~0.6 g). Then, $Ca_2Nb_3O_{10}$ nanosheets were exfoliated by shaking for 10 days in atmosphere. The residual TBAOH was washed away by centrifugation and washing with deionized water. Finally, the $Ca_2Nb_3O_{10}$ nanosheets were dispersed in deionized water by adding 100 mL of deionized water per 0.01 mol $Ca_2Nb_3O_{10}$ to obtain the nanosheets precursor solution (PS).

### Assembly of nanosheets films and device fabrication

Si and glass substrates were washed with acetone, anhydrous ethanol, and deionized water and then hydrophilized with plasma for 5 min. The D-CNO film was prepared by dropping 100 µL of the CNO dispersion onto the glass substrate (1.5 cm × 1.5 cm). To prepare S-CNO films, a dispersion was prepared according to the volume ratio of CNO PS: deionized water: anhydrous ethanol = 1:2:3. The substrates were placed on a heating tablet at 100 °C and the dispersion solution was loaded into a pneumatic spray gun for manual spraying. S-CNO 4, 6, 8, 10, 12 mL (i.e., sprayed dispersion = 24, 36, 48, 60, 72 mL) were obtained throughout the spraying process (designating the film prepared by spraying 4 mL of precursor solution as S-CNO 4 mL, and so on). To fabricate photodetectors, 100 nm Au electrodes were evaporated onto the films with a shadow mask (Fig. 3a). To construct the large-area flexible devices array, an Au electrode array with a thickness of 100 nm was firstly obtained using photolithography and thermal evaporation. S-CNO films were prepared onto the Au electrode array to construct the device array. More details can be found in Supplementary Fig. 22 and Supplementary Fig. 23. To fabricate the hole-only devices (ITO-PEDOT-CNO-MoO$_3$-Ag) for SCLC test, PEDOT solution was spin-coated on ITO substrate (3,000 rpm for 30 s) and the annealed at 130 °C for 15 min, followed by the preparation of D-CNO and S-CNO films. Then, 10 nm $MoO_3$ was thermal evaporated on the films, followed with the evaporation of 30 nm Ag.

### Material characterization and optoelectronic measurements

The optical photographs were taken by an Olympus optical microscope. The UV-vis absorption and transmittance spectra were obtained using a Hitachi U-4100 UV-vis spectrophotometer. The XRD patterns were obtained by a Bruker D8A25 equipped with Cu K$_\alpha$ radiation ($\lambda$ = 1.5405 Å). The GIWAXS patterns were carried out with glass substrate on beamline BL16B1 at the Shanghai Synchrotron Radiation Facility (SSRF) using X-rays with a wavelength of 1.2398 Å. The XPS data were collected by the PHI5000C & PHI5300 X-ray photoelectron spectrometer equipped with a dual Mg/Al anode. All peaks were calibrated with the C 1s peak (284.6 eV), and deconvolution was performed using a Gaussian-Lorentzian fitting after background subtraction. The PL emission spectrum and TRPL were characterized using Edinburgh FLS1000 with an EPL-375 nm laser. The surface morphologies of the CNO nanosheet films were characterized by field emission SEM (Zeiss Sigma) and AFM (Bruker Dimension Edge atomic force microscope). TEM images and energy dispersive X-ray spectroscopy (EDS) results were obtained using JEOL JEM-F200. The semiconductor test system Keithley 4200-SCS equipped with a monochromator and a light source system consisting of a continuous tunable 75 W Xe lamp was utilized for optoelectronic tests (such as *I-V* and *I-t* curves). The power density of the light was measured with an Ophir Photonics NOVA II optical power meter. A resistor, a Tektronix MSO/DPO5000 oscilloscope, and a transient light response system with a Q-switch YAG: Nd laser (Continuum Electro-Optics, MINILITE II, pulse duration: 20 µs, 355 nm) were used to collect the transient photoresponse data. The contact angles were measured using a contact angle analyzer (OCA25, Dataphysics, Germany), and the average values were adopted from at least three parallel measurements using a 2 µL droplet. The flexible tests were performed on the flexible electronic tester (FT2000). Note that all the box plots in the manuscript contain the median line, mean value, outlier that lies beyond the whiskers, box limit from the lower quartile to the upper quartile, and whiskers that extend from the box to the smallest and larges values within 1.5 times the IQR from the first and third quartiles.

### Designing and training of the convolutional neural network

The structure of the designed convolutional neural network is shown in Fig. 5b. The CNN model consists of four convolutional layers, two max-pooling layers, and a fully connected layer, and the photocurrent matrix is programed to represent the convolution kernel to preprocess the image in the optoelectronic domain. A loss function was defined to quantify the performance of our model on images with known tags and was used to train the CNN. It is the percentage of inaccuracy between the predicted and observed markers. We employed a particular form known as Categorical for classification (> 2 classes). The RMSPROP update makes minor adjustments to the ADAGRAD method to lessen its aggressive and monotonically decreasing learning rate. Based on the data generated by the sensor array, a tailor-made data set consisting of eight types of motion (left to right, right to left, up to down, down to up, left-up to right-down, right-down to left-up, left-down to right-up and right up to left-down) was developed for training and testing. The learning rate (LR) annealing approach was utilized to make the optimizer converge more quickly and more closely to the global minimum of the loss function. The optimizer navigates across the loss landscape with LR. Larger steps and faster convergence are associated with greater LR. Nevertheless, the optimizer may encounter a local minimum due to the low sample quality and high learning rate. To effectively approach the global minimum of the loss function during training, it is preferable to have a decreasing learning rate. To maintain the benefit of a high learning rate in fast computation time, the learning rate will be halved if the accuracy does not increase after three epochs. After training 30 iterations in one epoch with a batch size of 86, the loss of train and test sets converged to a steady level, verifying that the CNN was well-trained and did not suffer from under- and over-fitting problems.

## Data availability

The data that support the findings of this work are provided in the main text and the Supplementary Information. More relevant data are available from the corresponding authors upon request. Source data are provided with this paper.

## Code availability

The codes and dataset that used for designing and training the convolutional neural network are available from the corresponding authors upon request.

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

## Acknowledgements

This work is supported by National Natural Science Foundation of China (No. 62374035 (X.S.F.), 92263106 (X.S.F.), 62204047 (Z.Q.L.), 52425308 (X.S.F.), and 12211530438 (X.S.F.)). Thanks to Xinyu Zhang for the great assistance in conducting photolithography. Thanks to Zijun Hu for performing the SEM characterizations and taking photos in Fig. 4. Thanks to Tingting Yan for performing the AFM characterizations. Thanks to Zijin Zhao for helping with the GIWAXS test.

## Author contributions

M.D., X.S.F., Z.Q.L., and L.M.W. conceived the idea and designed the experiments. M.D. conducted the experimental work, performed measurements and analysis, and composed the manuscript under the supervision of X.S.F., Z.Q.L., and L.M.W. S.Y.L. designed and trained the CNN model, generated the dataset, and assisted in writing the manuscript. All authors contributed to the work and commented on the paper.

## Competing interests

The authors declare no competing interests.
