## [Peer Review File · Nature Communications]

Wafer-scale integration of two-dimensional perovskite oxides towards motion recognitionREVIEWER COMMENTS

Reviewer #1 (Remarks to the Author):

The integration of two-dimensional materials is an important topic in the optoelectronic field, and the large-area integration of two-dimensional perovskite oxides remain a challenge. In this manuscript, the authors proposed a charge-assisted oriented assembly film-formation process to fabricate wafer scale thin film. Furthermore, the authors demonstrated the applications for flexible electronics. They also adopt the devices to demonstrate towards motion recognition. The design is novel and the results are sufficiently supportive for the conclusions. I would like to recommend the acceptance for the publication in Nature Communications, provided if the authors can address the following questions properly.

1. As we know, the nanosheets are usually electroneutral and have a balanced charge. The authors point out that a charge-assisted oriented assembly film-formation process is introduced. How can authors introduce the charge on the prepared nanosheets? Or how the charged nanosheets maintain the charge balance in the solution? Some explanation should be enriched more clearly.

2. According to the experimental data in Figure 1d, 3e, the solution amounts play an important effect on the quality of the as-prepared thin film. The authors indicate that there are differences in the optoelectronic performances of the thin films prepared by different amounts of solutions. Please provide more experimental evidences and reasonable explanation to support the corresponding microstructures.

3. Some points should be further clarified towards the film preparation on the flexible substrate. How is the infiltration between the target solution and flexible Parylene-C during the manufacturing process? The large-area uniform film can be obtained on the silicon wafer, how can the author solve the problem of film formation and film uniformity on the Parylene-C?

4. The as-constructed flexible devices show good uniformity. However, their photocurrents are lower than those of the rigid devices. Please explain the reasons and discuss some solution strategy. Moreover, it is suggested to provide the impact of the flexibility test on the optoelectronic performances.

5. In the motion recognition, the authors choose 16 x 16 array devices. Is there any impact of the integration degree on the intensive reading of the trajectory detection? For example, how about 8 x 8 or 20 x 20, which one has a better accuracy? The author needs to explain the selection principle of the array scale. How is the device variation? Can this device be used for the event-driven vision sensors (Nature Electronics, 2023, 6, 870-878)?

Reviewer #2 (Remarks to the Author):

The authors reported a charge-assisted oriented assembly film-formation process to realize large-area ordered orientation of nanosheet functional units. They achieved recognition of multidirectional motion trajectories with a high accuracy of over 99.8%, presenting insights into the large-area fabrication and utilization of 2D perovskite oxides. The manuscript is well written and the work is relatively solid. However, I recommend the publication of this manuscript after major revision addressing the following issues:

1. In figure 1a, the authors proposed the COAF strategy to form uniform nanosheets. If nanosheets get ordered by repulsion, how can they fill in the blank space between the nanosheets already sedimented? The author's schematic illustration cannot fully explain the film formation without any voids.

2. Although the nanosheets are uniformly oriented due to repulsion and compactly packed in the out-of-plane direction, the mechanism in the in-plane direction remains unclear. Moreover, the presence of any empty spaces in the in-plane direction could weaken the advantages, especially considering that the devices discussed in this manuscript have a planar structure.
3. The size of nanosheets in S-CNO appears to be smaller than those in D-CNO films. Is it possible that the smoother surface of S-CNO films is due to the smaller size of their constituent components?
4. While the authors have demonstrated S-CNO films with significantly improved smoothness, it should be noted that enhanced smoothness does not necessarily translate to fewer defects or improved carrier transport. It is recommended to include quantitative analyses based on experimental results to strengthen the findings presented in this manuscript.
5. In the XRD pattern, S-CNO exhibits leftward shifts in the diffraction peaks compared to the D-CNO film, and the authors attribute this to the insertion of large TBA cations. However, if TBA cations indeed affect the crystal interplanar spacing, one would expect to observe peak shifts, potentially even larger ones, in the high-angle region as well. It is unclear why leftward shifts occur only in the low-angle region.
6. To prove reduced surface tension for better dispersion, it would be better to add contact angle image according to each concentration.
7. The authors mentioned S-CNO 10ml is optimal condition. Based on optical image according to the concentration of the S-CNO, Fig.S3, Fig.S4, Fig.S19, some impurities look like white particle (Fig.S19) are increased as the concentration increases. It may have bad effect on the device properties, what is it?
8. To obtain the photo-parameter (responsivity, detectivity,...), the area of electrode is needed. Please add it to the Fig.3a.
9. Please check the unit of scale in fig.3i. The scale is usually represented with area. Do all 2D perovskite oxide-based film photodetectors in figure.3i have square shape? If so, it is reasonable.
10. It would be better to add 30° bending condition in Fig.S21
11. In figure 4, the authors used parylene-C film to make flexible substrate. It should be addressed how much the substrate can be bended while maintaining the performance of the device. Also in figure 4d, photocurrent looks quite decreased relative to the photocurrent of figure 3b. The photocurrent seems to be decreased by 3 order.
12. Providing more information about data processing would enhance the understanding of the manuscript. Additional explanations regarding flexible scenarios (Fig. 4f), transmittance wavelength and absorbance normalization conditions (Supplementary Fig. 9), and definitions of bending angle and the methodology used for measuring device performance (Supplementary Fig. 21) would improve the quality of this manuscript.
13. In part 4, the authors should add the supplementary video of the device performing motion recognition. In figure 5, the authors used the terminology “array”. Although the device is working well for falling straw hat, the authors should show that the motion recognition works well with multiple distinct motion; at least two in the way author shows the device sensing the motion in figure 5c to say the device as array.

Reviewer #3 (Remarks to the Author):

Deng et al. demonstrated a COAF process to prepare wafer scale oxide perovskite devices for fast response, showing better accuracy in motion recognition compared to the control

group. However, the reported response times ($t_r = (42.4 \pm 14.8) \mu\text{s}$ and $t_d = (1.77 \pm 0.40) \text{ms}$) significantly lag behind state-of-the-art perovskite devices, such as $t_r/t_d = 0.3/0.5 \mu\text{s}$ as documented in Nature Communications, 2024, 15, 2066 (<https://doi.org/10.1038/s41467-024-46468-5>). Therefore, the publication in Nature Communications is not recommended. Here are some specific comments intended to aid in refining the manuscript for the publication in another Journal.

1. The authors claim that “the surfaces of the S-CNO films are significantly smoother, indicating fewer defects...”. Whether there is a quantitative relationship between surface roughness and the number of defects needs to be supported by experimental data.
2. The surface of the mainstream two-dimensional material usually has van der Waals characteristics, with relatively low dangling bonds. The non-van der Waals characteristics of oxide perovskite nanosheets make them have high-density dangling bonds, which has an impact on the performance of the device ? Does the residual solvent in the solution treatment process lead to the degradation of device performance ?
3. Compared with the control group, the optimized device has higher recognition accuracy in the Motion recognition task. However, only comparing the devices prepared in their own two states cannot effectively reflect the advanced nature of the devices. It needs to be compared with commercial devices, such as silicon devices, GaAs, etc. for the same task.
4. Although the authors claim that the nanosheets have good orientation, the morphology of the nanosheets photographed by SEM is irregular (Fig.1c i-iii), and it seems that the nanosheets are stacked in the film. In this case, there is a large amount of lattice mismatch at the contact interface between the nanosheets. Will it cause charge capture to produce built-in electric field and other effects ? For example, in Fig.3g, under certain illumination conditions, the voltage corresponding to the minimum current of the device is not 0, which means that there is a built-in electric field. Please discuss.
5. Some pictures need to be corrected or further discussed.
 - a. Fig 3b needs to add errorbar
 - b. Fig.3i suggests increasing the comparison of halide perovskite and other two-dimensional material categories.
 - c. Fig 3g shows why the voltage value corresponding to the minimum current value under different light conditions is different (e.g., from bottom to top line 3-5).

Response to the reviewers' comments

Reviewer #1

The integration of two-dimensional materials is an important topic in the optoelectronic field, and the large-area integration of two-dimensional perovskite oxides remain a challenge. In this manuscript, the authors proposed a charge-assisted oriented assembly film-formation process to fabricate wafer scale thin film. Furthermore, the authors demonstrated the applications for flexible electronics. They also adopt the devices to demonstrate towards motion recognition. The design is novel and the results are sufficiently supportive for the conclusions. I would like to recommend the acceptance for the publication in Nature Communications, provided if the authors can address the following questions properly.

Response: We thank you very much for the positive comments on the significance and innovations of our work. Your valuable comments and suggestions have really helped us to refine it. According to the comments, we have carefully revised our manuscript. The revised parts are marked in red in the revised manuscript. Our point-to-point responses are listed below.

Comment 1. *As we know, the nanosheets are usually electroneutral and have a balanced charge. The authors point out that a charge-assisted oriented assembly film-formation process is introduced. How can authors introduce the charge on the prepared nanosheets? Or how the charged nanosheets maintain the charge balance in the solution? Some explanation should be enriched more clearly.*

Response: We appreciate you for raising this question and making us recognize the incompleteness of the related interpretation. During the synthesis of CNO nanosheets, the acid converts $\text{KCa}_2\text{Nb}_3\text{O}_{10}$ to $\text{HCa}_2\text{Nb}_3\text{O}_{10}$, then the protons in $\text{HCa}_2\text{Nb}_3\text{O}_{10}$ are replaced by TBAOH treatment to achieve exfoliation, and finally TBA^+ cations are washed away to produce $\text{Ca}_2\text{Nb}_3\text{O}_{10}^-$ nanosheets. In the above process, negative charges are introduced into the nanosheets due to the stripping of interlayer cations¹⁻³. And to better prove the existence of repulsive forces between the nanosheets, we complemented the *Supplementary Information* with the uniform dispersion of the nanosheets in aqueous solution after a period of placement (as shown in Fig. R1).

Fig. R1 | Photographs of CNO nanosheets in aqueous solution. (a) control pure water, (b) fresh CNO nanosheets solution, (c) after placement for 1 week, (d) after placement for 1 month, (e) after placement for 3 months, (f) after placement for 6 months.

In our revised manuscript, the corresponding changes are listed as follows:

- ① On the 91st line of page 5, the sentence “**Given the special intrinsic negative charge of CNO (Supplementary Fig. 2),** to achieve the large-area homogeneous preparation, we **further** introduce a charge-assisted oriented assembly film-formation (COAF) strategy (~~see Methods section for more details~~)”.
- ② On the 665th line of page S3, related discussions have been added to the caption of Supplementary Fig. 1: “**Note that after acid converts $KCa_2Nb_3O_{10}$ to $HCa_2Nb_3O_{10}$, the protons in $HCa_2Nb_3O_{10}$ are then replaced by TBAOH treatment to achieve exfoliation, and finally TBA^+ cations are washed away to produce $Ca_2Nb_3O_{10}^-$ nanosheets. In the above process, negative charges are introduced into the nanosheets due to the stripping of interlayer cations, which is proved in **Supplementary Fig. 2**”.**
- ③ On the 671st line of page S3, **Supplementary Fig. 2** has been added and related discussions have been added to its caption: “**After several months of placement, the nanosheets remains dispersed in solution without obvious settling or aggregation, proving the existence of repulsive forces between the nanosheets**”.

Comment 2. *According to the experimental data in Figure 1d, 3e, the solution amounts play an important effect on the quality of the as-prepared thin film. The authors indicate that there are differences in the optoelectronic performances of the thin films prepared by different amounts of solutions. Please provide more experimental evidences and reasonable explanation to support the corresponding microstructures.*

Response: We appreciate you for the constructive comment. The amount of solution

affects the optoelectronic performance of the device mainly through affecting the thickness of the film. That is, the more the amount of precursor solution is used, the thicker the film forms. As we explained in detail in Supplementary Note II, when the thickness of the film is too thin, less effective absorption of light results in the generation of fewer photogenerated carriers, leading to a lower device photocurrent. And as the film thickness increases, the enhancement in light absorption leads to an increase in device performance, resulting in the best performing S-CNO 10 ml device. However, with further increase in film thickness, the photogenerated carriers need to travel longer distances in the film to be collected by the electrodes. Due to defects inside the film, the carriers may recombine in the process and fail to reach the electrodes, ultimately leading to the decrease in performance. Therefore, as the amount of precursor solution increases, the devices exhibited a process of increasing and then decreasing performance, as shown in Fig. R2.

Fig. R2 | **a-e** Semilogarithmic I - V curves under 280 nm, and **f-j** I - t curves under 280 nm and 5 V bias of S-CNO film devices with different thicknesses. **(a)** and **(f)** S-CNO 4 ml. **(b)** and **(g)** S-CNO 6 ml. **(c)** and **(h)** S-CNO 8 ml. **(d)** and **(i)** S-CNO 10 ml. **(e)** and **(j)** S-CNO 12 ml.

Comment 3. *Some points should be further clarified towards the film preparation on the flexible substrate. How is the infiltration between the target solution and flexible Parylene-C during the manufacturing process? The large-area uniform film can be obtained on the silicon wafer; how can the author solve the problem of film formation and film uniformity on the Parylene-C?*

Response: We thank you for raising the valuable questions and respond to each of the

two questions:

(1) For the infiltration issue, as we described in the **Methods** section, we applied a dispersion with the volume ratio of CNO precursor solution: deionized water: anhydrous ethanol = 1:2:3 by adding ethanol into the CNO precursor solution for the spray-coating process. We believe that the addition of ethanol not only reduces the surface tension for better dispersion of nanosheets, but also improves the infiltration between solution and Parylene-C film. To illustrate this, we tested the contact angles of the solvent mixture droplets on both glass and Parylene-C (Fig. R3). It can be seen that the contact angles decrease with increasing ethanol concentration on both glass and Parylene-C, indicating that the increasing the concentration of ethanol reduces the surface energy of the solvent and improves its wettability with the substrates.

Fig. R3 | Contact angle images of solvent mixture droplets (CNO precursor solution, deionized water and anhydrous ethanol) on glass (**a-d**) and Parylene-C (**e-h**) with volume ratio of water: ethanol=2:0 (**a** and **e**), water: ethanol=2:1 (**b** and **f**), water: ethanol=2:2 (**c** and **g**), water: ethanol=2:3 (**d** and **h**).

(2) For the uniformity issue of the film on the flexible Parylene-C, theoretically, since one of the most significant advantages of the spray-coating method we used is the adaptability to the substrates, and since the homogeneity of the films mainly depends on our proposed COAF process, we believe that replacing the silicon wafer with the rigid substrate with a flexible Parylene-C substrate basically has few effects on the film homogeneity. Meanwhile, from the practical experimental data, as shown in Fig. 4d and 4f, the photocurrents uniformity of all devices on Parylene also verifies the film uniformity.

In our revised manuscript, the corresponding changes are listed as follows:

① On the 100th line of page 5, the sentence “The solvent mixture of deionized water and ethanol not only provides reduced surface ~~tension~~ energy for better dispersion of solutes and enhanced wettability of the solution on substrates (Supplementary Fig. 4), but also enables the nanosheets to instant stack layer-by-layer with the rapid volatilization of solvent to avoid the percolation issue”.

② On the 487th line of page 24, the sentence “The contact angles were measured using a contact angle analyzer (OCA25, Dataphysics, Germany), and the average values were adopted from at least three parallel measurements using a 2 μ L droplet”.

③ On the 682nd line of page S4, Supplementary Fig. 4 has been added and related discussions have been added to its caption: “As the concentration of ethanol increases, the contact angles of the solvent mixture droplets on both glass and Parylene-C decrease, suggesting that the increase in ethanol reduces the surface energy of the solvent and improves its wettability with the substrates”.

Comment 4. *The as-constructed flexible devices show good uniformity. However, their photocurrents are lower than those of the rigid devices. Please explain the reasons and discuss some solution strategy. Moreover, it is suggested to provide the impact of the flexibility test on the optoelectronic performances.*

Response: We thank you for the suggestion and respond to each of the two questions:

(1) For a significant issue, i.e., the seeming decrease in the photocurrent of devices on flexible Parylene-C substrate versus that on rigid substrate, there are two main reasons:

a. Difference in the device channel size (i.e., effective illuminated area); **b. Difference in the applied bias.**

Specifically, the photocurrent of the rigid devices in Fig. 3b and 3c were measured with an effective illuminated area of 50000 μm^2 and an applied bias of 5 V, while the photocurrent of the flexible devices in Fig. 4d and 4f were measured under condition of effective illuminated area of 4500 μm^2 and applied bias of 1 V. As for the reasons for reduce the channel area and bias, smaller channel area can increase the integration degree of the device array, and for practical application-oriented devices, smaller bias means lower energy consumption while ensuring functionality. Therefore, the

reduction in illuminated area and bias results in a reduction in photocurrent. But this does not mean that flexible devices perform significantly worse. To prove this, we calculate the R_{1V} (responsivity under 1 V bias) for the rigid device at 1 V bias by extracting the data from revised Fig. 3h, and find that the performance of the rigid/flexible devices at the same conditions is at the same level, as shown below,

$$R_{\text{rigid},1V,280\text{nm}} = \frac{I_p - I_d}{P_\lambda \cdot S} = \frac{2.99 \times 10^{-7} \text{A} - 3.14 \times 10^{-13} \text{A}}{703 \mu\text{W}/\text{cm}^2 \times 50000 \mu\text{m}^2} = 0.85 \text{ A/W}$$

$$R_{\text{flexible},1V,280\text{nm}} = \frac{I_p - I_d}{P_\lambda \cdot S} = \frac{9.08 \times 10^{-9} \text{A} - 5.24 \times 10^{-12} \text{A}}{703 \mu\text{W}/\text{cm}^2 \times 4500 \mu\text{m}^2} = 0.29 \text{ A/W}$$

Note that the data used above are taken from the original manuscript as shown in Fig R4:

Fig. R4 | (a) revised Fig. 3h (original Fig. 3g), (b) Fig. 4d and (c) revised Fig. S24c (original Fig. S20c) in the revised manuscript. Note that (a) represents the data of rigid device, and (b) and (c) represent the data of flexible device.

Therefore, calculated data show that **the lower photocurrents of the flexible devices are rational and does not mean a performance degradation**, and can be improved by increasing the applied bias. However, we have to point out that we made a shameful mistake in the caption of Fig. 3b, where the “5 V bias” was misspelled as “1 V bias”, and we have made the correction (comparing the data in Fig. 3b and Fig. 3h reveals that this was an unintentional mistake). Meanwhile, we have complemented the channel sizes and applied bias for both rigid and flexible devices in the caption of Fig. 3 and Fig. 4, as well as the corresponding contents. And we have also clarified in the revised manuscript that “Note that the relatively low photocurrents stem from the reduction in device channel and applied bias and this does not mean the significant performance decline compared with rigid device

(Supplementary Fig. 24)”.

(2) Then, based on the relevant comments of you and reviewer #2, we have refined the flexibility test and depicted its impact on device performance, as shown in Fig. R5. And corresponding modification are presented in the revised manuscript.

Fig. R5 | **a**, Semilogarithmic $I-t$ curves measured at different bending angles under 1 V bias and 280 nm UV illumination. **b**, Optical image of the flexible device on a flexible electronic tester. **c**, Optical images of the bending test at different bending angles of 0°, 60°, 90°, 120°, 150° and 180°. **d**, Photocurrents of the device under bending angles from 0° to 180°.

In our revised manuscript, the corresponding changes are listed as follows:

- ① On the 223rd line of page 11, the sentence “**With a channel area of 50000 μm^2** , as shown in Fig. 3c, the optimal S-CNO 10 ml photodetector achieves a prime responsivity and a remarkable detectivity of 11.9 A/W and 3.71×10^{14} Jones under 280 nm UV illumination **and 5 V bias**”.
- ② On the 293rd line of page 14, the sentence “Photocurrents and on-off ratios of S-CNO devices measured under **+ 5 V bias** and 280 nm UV illumination”.
- ③ On the 319th line of page 15, the sentence “The overall structure of the active area

is schematically shown in Fig. 4a, and the inset shows a channel area of $4500 \mu\text{m}^2$ for each individual device”.

④ On the 326th line of page 15, the sentence “...with a low variation coefficient (C_V) of 17.6% under 280 nm illumination and 1 V bias, as shown in Fig. 4d and Supplementary Fig. 24. Note that the relatively low photocurrents stem from the reduction in device channel and applied bias and this does not mean the significant performance decline compared with rigid device (Supplementary Fig. 24)”.

⑤ On the 347th line of page 17, the sentence “The inset shows an optical microscopy image of an individual S-CNO device with a pair of interdigital electrodes and a channel area of $4500 \mu\text{m}^2$ ”.

⑥ On the 489th line of page 24, the sentence “The flexible tests were performed on the flexible electronic tester (FT2000)”.

⑦ On the 988th line of page S22, Supplementary Fig. 24 has been updated and related discussions have been added to its caption: “Note that, while exhibiting an average photocurrent of $(9.08 \pm 1.60) \times 10^{-9}$ A and an average dark current of $(5.24 \pm 6.28) \times 10^{-12}$ A, these flexible devices reveal an average responsivity of 0.29 A/W under 1 V bias and 280 nm illumination, which is very close to the that of rigid device (0.85 A/W) under the same test conditions”.

⑧ On the 996th line of page S23, Supplementary Fig. 25 has been supplemented as shown below and related discussions have been added to its caption: “It can be seen that significant performance degradation of the device occurs when the bending angle is larger than 150° ”.

Comment 5. *In the motion recognition, the authors choose 16 x 16 array devices. Is there any impact of the integration degree on the intensive reading of the trajectory detection? For example, how about 8 x 8 or 20 x 20, which one has a better accuracy? The author needs to explain the selection principle of the array scale. How is the device variation? Can this device be used for the event-driven vision sensors (Nature Electronics, 2023, 6, 870-878)?*

Response: We thank you for raising these questions and answer the above questions in three points:

(1) We accomplished the motion recognition function based on the as-constructed 16×16 devices array. This is because we believe that 256 pixels are enough to realize the designed motion recognition and show the performance differences between the experimental and the control groups, and of course the experiments proved so. So all the design, including the data acquisition of 256 pixels, the construction of neuromorphic network, and the data processing are based on this device. This is the principle on which we chose this integration degree.

(2) Regarding the effect of integration degree on the accuracy, we do believe that the higher the integration (i.e., the more pixels or the higher pixel density), the better. More pixels are beneficial for capturing more detailed information when the object is moving, and can also generate more valuable information for CNN training. Of course, we can implement device with higher integration degree by designing electrodes and increasing the number of pixels, but as we said above, the chosen 16×16 device array

is mainly intended to highlight the differences between the experimental and control devices. We believe that higher integration is indeed beneficial to optimize the motion recognition function, but may not be very beneficial to our innovation.

(3) To the best of our knowledge, the event-driven vision sensor presented in *Nature Electronics* function based on the ingenious and special device design and resulted unique property of producing tunable electrical signals only when the light intensity changes. Our photoconductive device cannot exhibit such functions and properties. But through the further design of the device structure, it is possible that this type of material can give similar devices richer choices of functionalities.

References

1. Kweon, S.-H. et al. Electrophoretic deposition of $\text{Ca}_2\text{Nb}_3\text{O}_{10}^-$ nanosheets synthesized by soft-chemical exfoliation. *J. Mater. Chem. C* **4**, 178-184 (2016).
2. Taniguchi, T. et al. Tunable Chemical Coupling in Two-Dimensional van der Waals Electrostatic Heterostructures. *ACS Nano* **13**, 11214-11223 (2019).
3. Xu, F., Ebina, Y., Bando, Y. & Sasaki, T. Structural characterization of (TBA, H) $\text{Ca}_2\text{Nb}_3\text{O}_{10}$ nanosheets formed by delamination of a precursor-layered perovskite. *J. Phys. Chem. B* **107**, 9638-9645 (2003).

Reviewer #2

The authors reported a charge-assisted oriented assembly film-formation process to realize large-area ordered orientation of nanosheet functional units. They achieved recognition of multidirectional motion trajectories with a high accuracy of over 99.8%, presenting insights into the large-area fabrication and utilization of 2D perovskite oxides. The manuscript is well written and the work is relatively solid. However, I recommend the publication of this manuscript after major revision addressing the following issues.

Response: We thank you very much for the positive comments on our manuscript. Your constructive and valuable comments and suggestions significantly help us to improve our work. According to the comments, we have carried out additional experiments and analysis, and have revised our manuscript accordingly. The modified parts are marked in red in the revised manuscript. Our point-to-point responses are listed below.

Comment 1. *In figure 1a, the authors proposed the COAF strategy to form uniform nanosheets. If nanosheets get ordered by repulsion, how can they fill in the blank space between the nanosheets already sedimented? The author's schematic illustration cannot fully explain the film formation without any voids.*

Response: We appreciate you raising this question that allows us to refine our corresponding interpretations. The COAF strategy in Fig. 1a focuses on the ordered arrangement of the main body, i.e. the larger nanosheets, in the formed film. But in fact, the prepared nanosheets solution contains nanosheets with lateral size ranging from a few micrometers (larger ones) to tens of nanometers (tinier ones). As it can also be seen from the practical experimental results in Supplementary Fig. 7 and 8, there seems to be no obvious holes on the surfaces of formed films. We believe that it is the tinier nanosheets that effectively filled the voids, and there are two main forces of the filling: **a. the same repulsion between the nanosheets, b. the percolation effect of the solvent accompanied by volatilization.**

More specifically, as Fig. R6 shows, when the larger nanosheets settle down, the tinier nanosheets will enter the blank space formed between the larger ones as the

solvent evaporates and percolates. During this process, they tend not to directly settle on the surface of the larger ones due to the repulsion, and undergo a similar COAF process inside the voids during the solvent evaporation, then may also form a roughly ordered stacking. Finally, they fill up these voids, forming films without obvious voids as shown in Fig. S7. In our original manuscript, we have actually made an explanation in Supplementary Note I-(II)-(iv): “Smaller nanosheets fill in the gaps as the solvent penetrates, and larger nanosheets gradually settle...”. But your question made us realize that this is too simplistic. To better illustrate this point, we further complemented corresponding schematic diagrams and interpretations in the main text and supplementary information. Finally, given the irregular shape of the nanosheets, we cannot deny that the interior structures of the formed films are totally void-free. But compared with the conventional D-CNO film, we believe that the advantage of our S-CNO films is that significant improvements in alignment and morphology have still been achieved.

Note that to highlight the main COAF process of larger nanosheets and the cleanliness of Fig. 1, we mainly supplemented the explanations in Supplementary Fig. 5. If you feel that this is not sufficient to illustrate the issue, please let us know and we will further modify the Fig. 1.

Fig. R6 | Complementary to the COAF process: void-filling process between large nanosheets. (1) Nanosheets with different sizes dispersed in the micro-droplet. (2) Larger nanosheets settle as the solution evaporates. (3) Tinier nanosheets enter the blank space formed between larger nanosheets as the solvent evaporates and percolates. (4) Tinier nanosheets fill in the blank spaces and voids to form continuous and uniform film.

In our revised manuscript, the corresponding changes are listed as follows:

① On the 112nd line of page 6, the sentence “As the solvent evaporates, the nanosheets in each micro-droplet settle and stack layer by layer, accompanied by the filling of voids and gaps between the larger nanosheets by the tinier ones

(Supplementary Fig. 5). But given the high shape irregularity of the nanosheets, the formed films may still not be totally void-free. After the complete evaporation of solvent, ~~forming~~ tight structures with high degree of orientation are formed (Fig. 1a, v)”).

② On the 699th line of page S5, **Supplementary Note. I** has been substantially complemented and refined: “(ii) **For each individual micro-droplet**, as the top-down exfoliation endows CNO nanosheets with intrinsic negative charges, the resultant repulsive forces prevent the aggregation of nanosheets, further contributing to their dispersion inside the **droplets**”.

③ On the 707th line of page S5, the sentence “(iv) **Since the prepared nanosheets precursor solution contains nanosheets with lateral size ranging from a few micrometers (larger ones) to tens of nanometers (tinier ones)**, it is also necessary to consider the behavior and significant role of tinier nanosheets in the COAF process – as the solvent evaporates and penetrates, they will ~~smaller nanosheets~~ fill in the gaps and voids to ensure the integrality of the films ~~as the solvent penetrates~~ (as details shown in **Supplementary Fig. 5**). Note that there are two main forces of the void-filing process: a. the same repulsion between the nanosheets (preventing tinier nanosheets from settling directly on the surface of larger nanosheets), b. the percolation effect of the solvent accompanied by volatilization. Meanwhile, the larger nanosheets gradually settle in this ordered arrangement as the solvent **dissipates volatilizes under heating**. However, considering the irregular shape of the nanosheets, there may still be some inevitable voids inside the nanosheets films.”.

④ On the 727th line of page S6, **Supplementary Fig. 5** has been added and related interpretations have been added to its caption.

Comment 2. *Although the nanosheets are uniformly oriented due to repulsion and compactly packed in the out-of-plane direction, the mechanism in the in-plane direction remains unclear. Moreover, the presence of any empty spaces in the in-plane direction could weaken the advantages, especially considering that the devices discussed in this manuscript have a planar structure.*

Response: We thank you for this comment and fully understand your concerns. In this regard, we will explain the in-plane arrangement of the nanosheets film from both microscopic and macroscopic perspectives. At the microscopic level, as described in the response to Comment 1, planar voids can be filled by tinier nanosheets, thus ensuring the in-plane spreading of the film in the horizontal direction and the planar migration of photo-induced carriers. At the macroscopic level, the in-plane arrangement is mainly dominated by the process. More specifically, a large number of ejected micro-droplets during the spraying process can complete the coverage of the entire surface of the substrate (Fig. R7a), allowing the nanosheets to be connected throughout the substrate plane. As shown in the revised Fig. 1b, at first a small number of droplets make the nanosheets distributed in isolated dispersion. But with the continuous replenishment of micro-droplets, the nanosheets will get connected to each other and form a continuous and uniform film (Fig. R7b), with surprising orientation and homogeneity throughout (Fig. 2d-f). Eventually, a structure with in-plane horizontal extension and out-of-plane stacking is formed.

Note that we have also made a corresponding supplementary note in the main text to complete our explanation of the COAF strategy. And still, due to the inherent shape irregularity of the nanosheets, there may still be voids in films and affect the carrier transport to some extent. But our present work shows that there has already been a significant improvement over the random arrangement in conventional dropped-coated films, and the planar device possess impressive performance.

Fig. R7 | Schematic illustration of (a) the entire substrate covered by a large number of sprayed micro-droplets and (b) the coverage of nanosheets in specific region of the substrate.

In our revised manuscript, the corresponding changes are listed as follows:

① On the 104th line of page 5, the sentence “As shown in Fig. 1a, in this process, the sprayed large number of micro-droplets first ensure the large-area in-plane spreading on the substrate (Fig. 1a, i). And the repulsive forces induced by the intrinsic negative charges between the nanosheets in micro-droplets prevent aggregation and facilitate their dispersion (Fig. 1a, ~~i~~ and ii)”.

② On the 112nd line of page 6, the sentence “As the solvent evaporates, the nanosheets in each micro-droplet settle and stack layer by layer, accompanied by the filling of voids and gaps between the larger nanosheets by the tinier ones (Supplementary Fig. 5). But given the high shape irregularity of the nanosheets, the formed films may still not be totally void-free. After the complete evaporation of solvent, ~~forming~~ tight structures with high degree of orientation are formed (Fig. 1a, v). Meanwhile, since the large number of micro-droplets cover the substrate surface uniformly over a large area during the spraying process, the nanosheets within each droplet will connect with those adjacent ones to form an integral whole. Ultimately, the ordered stack and the connections covering the whole substrate of the nanosheets simultaneously completes the construction of the highly oriented film (Fig. 1a, vi)”.

③ On the 696th line of page S4, Supplementary Note I has been supplemented and updated: “For processes ~~i-v~~ vi in Fig. 1a: (i) The large number of tiny micro-droplets ejected from the gun ensure a uniform distribution of nanosheets, and as they cover the surface of the substrate, they also ensure the large-area in-plane spreading parallel to the substrate”.

④ On the 718th line of page S5, the sentence “(v) After the solvent completely evaporates, the layer-by-layer stackings of nanosheets ~~form films with~~ in each individual micro-droplet ensure high degrees of orientation and tight structures, ~~and uniform surfaces~~. At the same time, due to the large-area and uniform coverage of the substrate surface by a large number of micro-droplets during the spray-coating process, the nanosheets inside the droplets will connect with the adjacent ones to form a block of thin film while settling, forming the large-area in-plane spreading of nanosheets parallel to the substrate surface. (vi) Ultimately, the ordered stack and the connections covering the entire substrate of the nanosheets simultaneously completes

the construction of the highly oriented film”.

⑤ On the 145th line of page 7, Fig. 1a (v) has been added and Fig. 1 has been updated as shown below. Fig. 1d has been moved to Supplementary Fig. 8.

Comment 3. *The size of nanosheets in S-CNO appears to be smaller than those in D-CNO films. Is it possible that the smoother surface of S-CNO films is due to the smaller size of their constituent components?*

Response: We thank you for raising this question. In fact, we used exactly the same CNO nanosheets precursor solution (synthesized at the same time in the same batch) for the preparation of both S-CNO and D-CNO films, so there should be no difference in the materials. We suggest two possible explanations for your question. Firstly, given the random settling process of the nanosheets in the D-CNO films, we tend to think that it is the difference in the film-formation process that leads to the difference in uniformity. Comparing the SEM images as shown in Fig. R8, there are large nanosheets in both D-CNO and S-CNO films. However, the nanosheets in S-CNO films tend to be buried inside and form uniform films (Fig. R8d), whereas that in D-CNO films are exposed on the surface (Fig. R8c), which ultimately contributes to the difference in their roughness. Secondly, with the above reason, since the drop-coated films also contain relatively more nanosheets than the spray-coated films (relatively more precursor solution per unit area is needed to prepare the D-CNO film), the larger quantities of large nanosheets will also appear more obvious. And the large nanosheets on the surface of the S-CNO film will appear to be more dispersed (Fig. R8b). We have supplemented corresponding interpretations in the revised manuscript. Mention that the reason for the creation of nanosheets with different size (larger or

tinier) is explained in the response to **Comment 7**.

Fig. R8 | SEM image of D-CNO film (a and c) and S-CNO film (b and d).

In our revised manuscript, the corresponding changes are listed as follows:

① On the 749th line of page S7, the sentence “**Note that large size nanosheets are observed on the surfaces of both D-CNO and S-CNO films, which are formed by incomplete exfoliation. And the nanosheets on the surface of D-CNO seem to be more and larger, which we believe is due to: (1) Difference in the film-formation process. Large nanosheets in S-CNO films tend to be buried inside and form uniform surfaces, while that in D-CNO films are exposed on the surfaces. (2) As relatively more precursor solution per unit area is needed to prepare the D-CNO film, there will be more large nanosheets in them than in S-CNO films, while the large nanosheets on the S-CNO films appear to be more dispersed**”.

Comment 4. *While the authors have demonstrated S-CNO films with significantly improved smoothness, it should be noted that enhanced smoothness does not necessarily translate to fewer defects or improved carrier transport. It is recommended to include quantitative analyses based on experimental results to strengthen the findings presented in this manuscript.*

Response: We appreciate you for the constructive suggestion.

Firstly, we believe that the improved smoothness of the S-CNO films is one of the possible reasons for the reduced defects and improved carrier transport, and more importantly, it is the orderliness of the nanosheets. Oriented stacking of nanosheets (shown in Fig. 2) implies the more ordered internal structure of the film, which is

certainly conducive to the reduction of defects and the improvement of carrier transport. To prove this viewpoint, we supplemented analyses to the TRPL test in Supplementary Fig. 13. Specifically, the decay process in TRPL test consists of a fast decay process (τ_1) and a slow decay process (τ_2), and τ_1/τ_2 are usually thought to be attributed to the surface/internal recombination and the shallow/deep-trap dominated trapping and de-trapping process, respectively¹⁻³. Compared with the slow decay of D-CNO film ($\tau_1 = 0.8682$ ns, $\tau_2 = 3.1264$ ns, $\tau_{ave} = 1.4802$ ns), the τ_1 , τ_2 , and τ_{ave} of S-CNO film ($\tau_1 = 0.9178$ ns, $\tau_2 = 3.7443$ ns, $\tau_{ave} = 1.6641$ ns) increased by 5.8%, 19.8%, and 12.4%, respectively. As longer average decay time τ_{ave} (i.e. longer carrier lifetime) represents better film quality and fewer defects and traps, these results demonstrate an overall reduction in traps, especially the more significant reduction in deep trap and internal defects inside the film, which supports our proposed perspective (revised Fig. 3g).

Furthermore, based on your constructive suggestion, we supplemented quantitative experiments by using the space-charge-limited current (SCLC) method to quantify the trap density⁴⁻⁶. Specifically, vertical hole-only devices (ITO-PEDOT-CNO-MoO₃-Ag) were fabricated with our strategy, and trap-filled limit voltages (V_{TFL}) were measured to calculate the hole-trap densities (n_{trap}), as shown in Fig. R9. And it can be seen that the n_{trap} of S-CNO devices reduced more than 10 times compared with that of D-CNO device. This not only verifies the fewer traps and defects, but also supports and corresponds to the interpretation presented in revised Fig. 3g.

More details in experiments and explanations are supplemented in our revised manuscript.

Fig. R9 | **a-e** Dark I - V characteristics of hole-only devices measured using the space-charge-limited current (SCLC) method. **(a)** S-CNO 6 ml device, **(b)** S-CNO 8 ml device, **(c)** S-CNO 10 ml device, **(d)** S-CNO 12 ml device, and **(e)** D-CNO device. **f**, Statistics of the V_{TFL} and n_{trap} calculated from every three D-CNO and S-CNO devices.

In our revised manuscript, the corresponding changes are listed as follows:

① On the 786th line of page S11, the sentence “The table shows the fitted parameters of the TRPL spectrum. Note that the decay process consists of two components, including a fast decay process (τ_1) and a slow decay process (τ_2). While τ_1 is usually thought to be attributed to the surface recombination and the shallow-trap dominated trapping/de-trapping process, τ_2 is related to the internal recombination and the deep-trap dominated trapping/de-trapping process¹⁻³. Compared with the slow decay of D-CNO film ($\tau_1 = 0.8682$ ns, $\tau_2 = 3.1264$ ns, $\tau_{ave} = 1.4802$ ns), the τ_1 , τ_2 , and τ_{ave} of S-CNO film ($\tau_1 = 0.9178$ ns, $\tau_2 = 3.7443$ ns, $\tau_{ave} = 1.6641$ ns) increased by 5.8%, 19.8%, and 12.4%, respectively. This demonstrates an overall reduction in traps, especially the significant reduction in deep trap and internal defects inside the film”.

② On the 251st line of page 12, the sentence “To provide a better illustration from a mechanistic point of view, we quantify the trap density (n_{trap}) by the space-charge-limited current (SCLC) method. As shown in Supplementary Fig. 19, the I - V curves of tailor-made hole-only S-CNO and D-CNO devices (ITO-PEDOT-CNO-MoO₃-Ag) can be divided into three typical SCLC regions, and

the n_{trap} of S-CNO devices that calculated from trap-filled limit voltages (V_{TFL}) is significantly lower than that of D-CNO device (Fig. 3f)”.

③ On the 275th line of page 13, the sentence “Experimental results and interpretations in Supplementary Fig. 13 also support this perspective”.

④ On the 459th line of page 23, the sentence “To fabricate the hole-only devices (ITO-PEDOT-CNO-MoO₃-Ag) for SCLC test, PEDOT solution was spin-coated on ITO substrate (3000 rpm for 30 s) and the annealed at 130 °C for 15 min, followed by the preparation of D-CNO and S-CNO films. Then, 10 nm MoO₃ was thermal evaporated on the films, followed with the evaporation of 30 nm Ag”.

⑤ On the 872nd line of page S15, the sentence “(III) SCLC test: As shown in Supplementary Fig. 19, the ohmic and trap-filled limit regions are distinctively separated at trap-filled limit voltage (V_{TFL}). The trap density (n_{trap}) can then be determined by the V_{TFL} with⁷⁻⁹

$$n_{trap} = \frac{2\epsilon_0\epsilon_r V_{TFL}}{eL^2}$$

where ϵ_0 denotes the vacuum permittivity of 8.85×10^{-12} F/m, ϵ_r is the dielectric constant of Ca₂Nb₃O₁₀ ($\epsilon_r = 26$)¹⁰, e represents the elementary charge of 1.6×10^{-19} C, and L is the film thickness”.

⑥ On the 291st line of page 14, Fig. 3f has been added and Fig. 3 has been updated as shown below. Former Fig. 3f and Fig. 3g have been moved, former Fig. 3h has been moved to Supplementary Fig. 14.

⑦ On the 892nd line of page S17, Supplementary Fig. 19 has been added.

Comment 5. *In the XRD pattern, S-CNO exhibits leftward shifts in the diffraction peaks compared to the D-CNO film, and the authors attribute this to the insertion of large TBA cations. However, if TBA cations indeed affect the crystal interplanar spacing, one would expect to observe peak shifts, potentially even larger ones, in the high-angle region as well. It is unclear why leftward shifts occur only in the low-angle region.*

Response: We thank you for this comment. First, we apologize for the inaccuracy of our description. We wanted to express that compared with HCNO, the peaks of the S-CNO and D-CNO films in the low-angle region are leftward shifted. Then, as for the reason why the leftward shifts occurred only in the low-angle region, we believe that your statement is correct and there are some issues with our explanation. For this observed experimental phenomena, after reviewing a large number of related publications, we did not find similar phenomena and reasonable interpretations. We speculate that this is possibly due to the special transition from 3D to 2D of this material, or the incomplete and superficial insertion of TBA⁺ in those thicker (multilayer) nanosheets. Because the uncertainty in interpretation and to avoid misleading readers, we have revised the corresponding explanation in our manuscript. And we can also complement our speculation in the manuscript if you feel that this issue is important.

In our revised manuscript, the corresponding changes are listed as follows:

① On the 158th line of page 8, the sentence “...illustrating that the nanosheets are all aligned along [001], as shown in Fig. 2a. ~~Note that the leftward shifts of the diffraction peaks in the low-angle region originate from the insertion of large TBA⁺ cations, as the increased interplanar spacing leads to a decrease in the diffraction angles.~~ Accordingly, we plot the micro-schematic of the cross-section...”.

Comment 6. *To prove reduced surface tension for better dispersion, it would be better to add contact angle image according to each concentration.*

Response: We appreciate the constructive advice. To prove the reduced surface tension, we tested the contact angles of the solvent mixture droplets on both glass and

Parylene-C, as shown in Fig. R10. It can be seen that the contact angles decrease significantly with increasing ethanol concentration on both glass and Parylene-C, indicating that the increasing the concentration of ethanol reduces the surface tension of the solvent and improves its wettability with the substrates.

Fig. R10 | Contact angle images of solvent mixture droplets (CNO precursor solution, deionized water and anhydrous ethanol) on glass (**a-d**) and Parylene-C (**e-h**) with volume ratio of water: ethanol=2:0 (**a** and **e**), water: ethanol=2:1 (**b** and **f**), water: ethanol=2:2 (**c** and **g**), water: ethanol=2:3 (**d** and **h**).

In our revised manuscript, the corresponding changes are listed as follows:

- ① On the 100th line of page 5, the sentence “The solvent mixture of deionized water and ethanol not only provides reduced surface ~~tension~~ **energy** for better dispersion of solutes **and enhanced wettability of the solution on substrates** (Supplementary Fig. 4), but also enables the nanosheets to instant stack layer-by-layer with the rapid volatilization of solvent to avoid the percolation issue”.
- ② On the 487th line of page 24, the sentence “**The contact angles were measured using a contact angle analyzer (OCA25, Dataphysics, Germany), and the average values were adopted from at least three parallel measurements using a 2 μ L droplet**”.
- ③ On the 682nd line of page S4, **Supplementary Fig. 4** has been added and related discussions have been added to its caption: “**As the concentration of ethanol increases, the contact angles of the solvent mixture droplets on both glass and Parylene-C decrease, suggesting that the increase in ethanol reduces the surface energy of the solvent and improves its wettability with the substrates**”.

Comment 7. *The authors mentioned S-CNO 10ml is optimal condition. Based on*

optical image according to the concentration of the S-CNO, Fig.S3, Fig.S4, Fig.S19, some impurities look like white particle (Fig.S19) are increased as the concentration increases. It may have bad effect on the device properties, what is it?

Response: We thank you for raising this question. An important step in the synthesis of CNO is the insertion of TBA⁺ into HCNO to promote the exfoliation of CNO nanosheets (Fig. S1). During this process, a tiny amount of thick nanosheets or bulk HCNO may remain in the solution due to the possible incomplete exfoliation, which is exactly the white particles in original Fig. S3, Fig. S4 and Fig. S19. But we think that in our work, whether they are impurities that impose a bad effect on the device properties needs to be further investigated. Because from the experimental results in Fig. 4d, it can be seen that in the presence of these particles, 256 devices can still achieve high electrical uniformity and consistent performance (the C_v of photocurrents is as low as 17.6%).

Meanwhile, we do understand your concern, as these relatively large nanosheets or particles inside the film may generate internal defects that affect carrier transport. To circumvent this problem, we figure out that by centrifuging the solution to some extent we can remove as much of these large particles as possible. But the process design for centrifugation still needs exploring, and this will be one of the research directions to optimize our future research. We have pointed out this problem in the revised main text and indicated the corresponding measures in the future.

In our revised manuscript, the corresponding changes are listed as follows:

① On the 139th line of page 7, the sentence “Incidentally, **these images still show some relatively large particles which are believed to be thicker nanosheets and incompletely exfoliated HC₂Nb₃O₁₀ (HCNO)**, and better optimization of film structure is expected to be realized by optimizing the synthesis...”.

Comment 8. *To obtain the photo-parameter (responsivity, detectivity,...), the area of electrode is needed. Please add it to the Fig.3a.*

Response: We thank you for the careful suggestion and have made corresponding supplement. Moreover, we also realized that the devices with different electrode areas

were not clearly labelled in the main text. For the sake of rigor, we have checked the main text and added the areas of electrodes to all the places in it where they should be provided.

In our revised manuscript, the corresponding changes are listed as follows:

- ① On the 222nd line of page 11, the sentence “A more specific interpretation is provided in Supplementary Note II. **With a channel area of 50000 μm^2** , as shown in Fig. 3c...”.
- ② On the 305th line of page 15, the sentence “**Note that the performance data in Fig. 3 and corresponding context are measure on D-CNO and S-CNO devices with channel areas of 50000 μm^2** ”.
- ③ On the 319th line of page 15, the sentence “The overall structure of the active area is schematically shown in Fig. 4a, **and the inset shows a channel area of 4500 μm^2 for each individual device**”.
- ④ On the 347th line of page 17, the sentence “The inset shows an optical microscopy image of an individual S-CNO device with a pair of interdigital electrodes **and a channel area of 4500 μm^2** ”.

Comment 9. *Please check the unit of scale in fig.3i. The scale is usually represented with area. Do all 2D perovskite oxide-based film photodetectors in figure.3i have square shape? If so, it is reasonable.*

Response: We thank you for this suggestion. We have modified the representation of “scale” from length to area and re-evaluated the area scale of all mentioned devices, as shown in Fig. R11. Incidentally, according to the suggestion of Reviewer 3, more comparisons of 2D halide perovskites and other 2D materials for UV photodetection have also been supplemented.

Fig. R11 | Modified performance comparison of representative 2D perovskite oxide-, 2D halide perovskite- and other 2D material-based film UV photodetectors, in which the data are concluded from Supplementary Table 2.

In our revised manuscript, the corresponding changes are listed as follows:

- ① On the 291st line of page 14, **Fig. 3i** has been updated as shown below and related statement has been added to its caption.

Comment 10. *It would be better to add 30° bending condition in Fig.S21*

Response: We thank you for the advice. Unfortunately, because subsequent bending tests irreversibly degraded the device performance (the film may split or cracked because of too much bending), we can no longer add the *I-t* curves for the same device in the revised Fig. S25 (i.e. original Fig. S21) at 30° bending condition. And we believe that if the device showed no degradation when being bent at a larger angle (like 60° as shown in Fig. S25), it is very likely that a smaller angle bending (30°) will have no effect. Therefore, given that the lack of this piece of data doesn't seem to

have a significant impact on our demonstration of the device flexibility, we earnestly request your understanding on this issue.

Comment 11. *In figure 4, the authors used parylene-C film to make flexible substrate. It should be addressed how much the substrate can be bended while maintaining the performance of the device. Also in figure 4d, photocurrent looks quite decreased relative to the photocurrent of figure 3b. The photocurrent seems to be decreased by 3 order.*

Response: We thank you for the constructive advice and will respond to the two questions separately:

(1) To figure out how much the substrate can be bent while maintaining the device performance, we re-prepared a device and supplemented the bending test with angles ranging from 0° to 180° as shown in Fig. R12. It can be seen that significant performance degradation of the device occurs when the bending angle is larger than 150°. And we have pointed this in the revised manuscript.

Fig. R12 | Photocurrents of the device under bending angles from 0° to 180°.

(2) For another significant issue, i.e., the seeming decrease in the photocurrent of devices on flexible Parylene-C substrate versus the photocurrent of devices on rigid substrate, there are two main reasons: **a. Difference in device channel size** (i.e., effective illuminated area); **b. Difference in the magnitude of the applied bias**.

Specifically, the photocurrent of the rigid devices in Fig. 3b and 3c were measured under an effective illuminated area of 50000 μm^2 and an applied bias of 5 V, while the photocurrent of the flexible devices in Fig. 4d and 4f were measured under condition of effective illuminated area of 4500 μm^2 and applied bias of 1 V. As for the reasons for reduce the channel area and bias, we believe that smaller channel area can increase

the integration degree of the device array, and that for practical application-oriented devices, smaller bias means lower energy consumption while ensuring functionality. Therefore, the reduction in illuminated area and bias results in a reduction in photocurrent. But this does not mean that flexible devices perform significantly worse. To prove this, we calculate the responsivity R_{1V} (responsivity under 1 V bias) for the rigid device at 1 V bias by extracting the data from the revised Fig. 3h, and find that the performance of the rigid/flexible devices at the same test conditions is at the same level, as shown below,

$$R_{\text{rigid},1V,280\text{nm}} = \frac{I_p - I_d}{P_\lambda \cdot S} = \frac{2.99 \times 10^{-7} \text{ A} - 3.14 \times 10^{-13} \text{ A}}{703 \mu\text{W}/\text{cm}^2 \times 50000 \mu\text{m}^2} = 0.85 \text{ A/W}$$

$$R_{\text{flexible},1V,280\text{nm}} = \frac{I_p - I_d}{P_\lambda \cdot S} = \frac{9.08 \times 10^{-9} \text{ A} - 5.24 \times 10^{-12} \text{ A}}{703 \mu\text{W}/\text{cm}^2 \times 4500 \mu\text{m}^2} = 0.29 \text{ A/W}$$

Note that the data used above are taken from the original manuscript as shown in Fig.

R13:

Fig. R13 | (a) revised Fig. 3h (original Fig. 3g), (b) Fig. 4d and (c) revised Fig. S24c (original Fig. S20c) in the revised manuscript. Note that (a) represents the data of rigid device, and (b) and (c) represent the data of flexible device.

Therefore, calculated data show that **the lower photocurrents of the flexible devices are rational and does not mean a performance degradation**, and can be improved by increasing the applied bias. However, we have to point out that we made a shameful mistake in the caption of Fig. 3b, where the “5 V bias” was misspelled as “1 V bias”, and we have made the correction (comparing the data in Fig. 3b and Fig. 3h reveals that this was an unintentional mistake). Meanwhile, we have complemented the channel sizes and applied bias for both rigid and flexible devices in the caption of Fig. 3 and Fig. 4, as well as the corresponding contents. And we have

also clarified in the revised manuscript that “Note that the relatively low photocurrents stem from the reduction in device channel and applied bias and this does not mean the significant performance decline compared with rigid device (Supplementary Fig. 24)”.

In our revised manuscript, the corresponding changes are listed as follows:

① On the 996th line of page S23, **Supplementary Fig. 25** has been updated and related discussions have been added to its caption: “**It can be seen that significant performance degradation of the device occurs when the bending angle is larger than 150°**”.

② On the 223rd line of page 11, the sentence “**With a channel area of 50000 μm^2** , as shown in Fig. 3c, the optimal S-CNO 10 ml photodetector achieves a prime responsivity and a remarkable detectivity of 11.9 A/W and 3.71×10^{14} Jones under 280 nm UV illumination **and 5 V bias**”.

③ On the 293rd line of page 14, the sentence “Photocurrents and on-off ratios of S-CNO devices measured under **+ 5 V bias** and 280 nm UV illumination”.

④ On the 319th line of page 15, the sentence “The overall structure of the active area is schematically shown in Fig. 4a, **and the inset shows a channel area of 4500 μm^2 for each individual device**”.

⑤ On the 326th line of page 15, the sentence “...with a low variation coefficient (C_V) of 17.6% under **280 nm illumination and 1 V bias**, as shown in Fig. 4d and Supplementary Fig. 20. **Note that the relatively low photocurrents stem from the reduction in device channel and applied bias and this does not mean the significant performance decline compared with rigid device (Supplementary Fig. 24)**”.

⑥ On the 347th line of page 17, the sentence “The inset shows an optical microscopy image of an individual S-CNO device with a pair of interdigital electrodes **and a channel area of 4500 μm^2** ”.

⑦ On the 988th line of page S22, **Supplementary Fig. 24** has been updated and related discussions have been added to its caption: “**Note that, while exhibiting an average photocurrent of $(9.08 \pm 1.60) \times 10^{-9}$ A and an average dark current of $(5.24 \pm 6.28) \times 10^{-12}$ A, these flexible devices reveal an average responsivity of 0.29 A/W**

under 1 V bias and 280 nm illumination, which is very close to the that of rigid device (0.85 A/W) under the same test conditions”.

Comment 12. *Providing more information about data processing would enhance the understanding of the manuscript. Additional explanations regarding flexible scenarios (Fig. 4f), transmittance wavelength and absorbance normalization conditions (Supplementary Fig. 9), and definitions of bending angle and the methodology used for measuring device performance (Supplementary Fig. 21) would improve the quality of this manuscript.*

Response: We appreciate your valuable suggestion and comment and have supplemented the corresponding information and discussion in the revised manuscript.

(1) In Fig. 4f, actually, we tested the photocurrents of the flexible device in the bent 90° state, which is shown in Fig. 4g. Additional explanations regarding flexible scenarios (Fig. 4f) have been complemented.

(2) In revised Fig. S12 (original Fig. S9), we did not normalize the absorbance spectrum and presented the raw and unprocessed data. We apologize for the mislabeling of the Y axis titles as “Normalized absorbance”. We have added the transmittance wavelength and made corresponding correction in the revised Fig. S12

(3) In revised Fig. S25 (original Fig. S21), we have complemented the corresponding definitions of bending angle, optical images and description for used methodology of the flexible tests, as shown in Fig. R14.

Fig. R14 | **a**, Semilogarithmic $I-t$ curves measured at different bending angles under 1 V bias and 280 nm UV illumination. **b**, Optical image of the flexible device on a flexible electronic tester. **c**, Optical images of the bending test at different bending angles of 0°, 60°, 90°, 120°, 150°, 180°.

In our revised manuscript, the corresponding changes are listed as follows:

- ① On the 335th line of page 16 (regarding Fig. 4f), the sentence “**Subsequently, the arrays are bent to 90°, and the $I-t$ curves of fifty pixels in the bending state are randomly measured. to display the functionality of the array in flexible scenarios (Fig. 4f).** As shown in Fig. 4f, the response behaviors of these pixels are essentially identical, and their photocurrents remain highly uniform with a C_v of 17.3%. It can be seen that morphology transformations do not interfere with the performance consistency, and **consequently** the device arrays **in the bending state** still show decent imaging result (Fig. 4g). Moreover, ultra-flexibility also allows the device array to conform to substrates of arbitrary shapes. By transferring the array onto a glass hemispherical substrate, a UV-sensitive retina-like device is presented (Fig. 4h), which **still also** exhibits clear and stable imaging capability (Fig. 4i).”
- ② On the 353rd line of page 17 (regarding Fig. 4f), the sentence “ $I-t$ characteristics of 50 pixel-units in the 90° bending state. **g**, Bending imaging results **under 90°**”.
- ③ On the 489th line of page 24, the sentence “**The flexible tests were performed on the flexible electronic tester (FT2000)**”.
- ④ On the 777th line of page S10, the sentence “The transmittance of S-CNO films

with different thicknesses tested under 280 nm UV”.

⑤ On the 776th line of page S10, Fig. S12b has been updated as shown below:

⑥ On the 996th line of page S23, Fig. S25 has been updated.

Comment 13. *In part 4, the authors should add the supplementary video of the device performing motion recognition. In figure 5, the authors used the terminology “array”. Although the device is working well for falling straw hat, the authors should show that the motion recognition works well with multiple distinct motion; at least two in the way author shows the device sensing the motion in figure 5c to say the device as array.*

Response: We thank you for the suggestion and comment.

Firstly, we think that in our work, the significance of execution process of motion recognition resides in the internal mechanism and flow, e.g. the process of pixel receiving stimulus and recovering (as shown in Fig. 5a, c and d). Therefore, after adequate discussions and pre-preparation, we concluded that the pictures were more illustrative and the video seemed to have limited enhancement to the presentation effect. And in actual execution, as we demonstrated that designing eight directions of motion is sufficient enough to show the advantages of the structural and performance

enhancements of our materials and devices. However, if we convert the process into a video, we are concerned that these processes, which are sophisticated in principle but relatively simple in representation, may instead appear somewhat monotonous. Therefore, we fully understand your concern, but it seems that the representation in Fig. 5 is sufficient to show the difference in effectiveness between the two group of devices, and the addition of video may not better highlight our innovation. But we do respect and value your opinion, and will provide an additional video is you insist that it is necessary.

Then, for the terminology “array”, we are following the concept as in Fig. 4, where we refer to the entire device consisting of the 16×16 pixel (photodetectors) as an array. Meanwhile, we need to clarify that we use Fig. 5a to illustrate the whole processing flow of our device to implement the motion recognition function, which is a schematic of the simulation and not the practical experimental result. We apologize for the misleading of our statement and have revised it accordingly. As we stated in the manuscript, the designed device and the corresponding neuromorphic networks can currently only achieve motion recognition in eight distinct directions. And these multiple distinct motions are shown in detail in the revised Fig. S28 and show decent recognition results. We believe that these designs are sufficient to demonstrate the potential of the device for the functionality, which is a result of our previous optimizations in device performance. Achieving recognitions of more complex motions (e.g. more complex trajectories, subjects with more complex geometries, etc.) relies on optimizing and further investigating the whole designed program, and may require increasing device integration degree (e.g. pixel count and density). We appreciate and value your significantly enlightening comments, and we will look into these perspectives in our future work.

In our revised manuscript, the corresponding changes are listed as follows:

① On the 371st line of page 18, the sentence “As the **schematic** processing flow in Fig. 5a shows, when the target object moves (**take a falling straw hat as an example**), the pixels in the sensor that receive the light...”

② On the 403rd line of page 20, the sentence “Schematic illustration of spatiotemporal vision imaging acquisition and motion recognition realization”.

References

1. Liu, X. et al. Enhanced response speed in 2D perovskite oxides-based photodetectors for UV imaging through surface/interface carrier-transport modulation. *ACS Appl. Mater. Interfaces* **14**, 48936-48947 (2022).
2. Mo, X. et al. Highly-efficient all-inorganic lead-free 1D CsCu₂I₃ single crystal for white-light emitting diodes and UV photodetection. *Nano Energy* **81**, 105570 (2021).
3. Wen, Z. et al. Graphene induced structure and doping level tuning of evaporated CsPbBr₃ on different substrates. *Chem. Eng. J.* **452**, 139243 (2023).
4. Dong, S. et al. All-inorganic perovskite single-crystal photoelectric anisotropy. *Adv. Mater.* **34**, 2204342 (2022).
5. Chen, J. et al. Single-crystal thin films of cesium lead bromide perovskite epitaxially grown on metal oxide perovskite (SrTiO₃). *J. Am. Chem. Soc.* **139**, 13525-13532 (2017).
6. Liu, Y. et al. A 1300 mm² ultrahigh-performance digital imaging assembly using high-quality perovskite single crystals. *Adv. Mater.* **30**, 1707314 (2018).

Reviewer #3

*Deng et al. demonstrated a COAF process to prepare wafer scale oxide perovskite devices for fast response, showing better accuracy in motion recognition compared to the control group. However, the reported response times ($t_r = (42.4 \pm 14.8) \mu\text{s}$ and $t_d = (1.77 \pm 0.40) \text{ms}$) significantly lag behind state-of-the-art perovskite devices, such as $t_r/t_d = 0.3/0.5 \mu\text{s}$ as documented in *Nature Communications*, 2024, 15, 2066 (<https://doi.org/10.1038/s41467-024-46468-5>). Therefore, the publication in *Nature Communications* is not recommended. Here are some specific comments intended to aid in refining the manuscript for the publication in another Journal.*

Response: We sincerely appreciate your comments and suggestions on our work, and we do understand your concerns on relatively slower speed compared with metal halide perovskites.

Firstly, it is well known that in the photodetection field, it always remains challenging to balance the high photosensitivity and the fast speed. And this contradiction is more common to 2D perovskite oxides^{1,2} - the new star in photodetection. While this material exhibits significantly higher photosensitivity compared with common metal halide perovskite (Supplementary Table 2), including the article (*Nature Communications*, 2024, 15, 2066) you mentioned, its speed still needs to be improved. Therefore, the innovation of our work lies not only in the first homogeneous large-area preparation of oxide perovskites, but also in the rare balance between photosensitivity and speed, which achieves the fastest response speed of any oxide perovskites currently available.

Secondly, as photodetectors used in different fields have different performance focuses, the to-be-improved response speed of our device does not represent the most important innovation of our work. For example, the work you mentioned (*Nature Communications*, 2024, 15, 2066) does achieve excellent response speed, but as it is designed towards optical communication applications, the device focuses only on speed advantages at the expense of photosensitivity. While in our work and in the vast majority of related work, photosensitivity is just as important as or more important than speed, and we need to consider the trade-off between multiple aspects of

performance.

Thank you again for your valuable suggestions which have helped us to improve our manuscript, and the revised parts are marked in red in the revised manuscript. Our point-to-point responses are listed below.

Comment 1. *The authors claim that “the surfaces of the S-CNO films are significantly smoother, indicating fewer defects...”. Whether there is a quantitative relationship between surface roughness and the number of defects needs to be supported by experimental data.*

Response: We appreciate your constructive suggestions. Given that the photodetection always imposes strict demand on high homogeneity and quality of photosensitive films³⁻⁵, we believe that the improved smoothness can be one of the possible reasons for the reduced defects. And the oriented stacking of nanosheets (shown in Fig. 2) is the more important reason which is certainly conducive to the reduction of defects, which is also verified by subsequent experiments on the improved response speed of devices. Based on your valuable suggestion and to better prove this, we have supplemented two quantitative analyses.

Firstly, we supplemented analyses to the TRPL test in Supplementary Fig. 13. Specifically, the decay process in TRPL test consists of a fast decay process (τ_1) and a slow decay process (τ_2), and τ_1/τ_2 are usually thought to be attributed to the surface/internal recombination and the shallow/deep-trap dominated trapping and de-trapping process, respectively⁶⁻⁸. Compared with the slow decay of D-CNO film ($\tau_1 = 0.8682$ ns, $\tau_2 = 3.1264$ ns, $\tau_{ave} = 1.4802$ ns), the τ_1 , τ_2 , and τ_{ave} of S-CNO film ($\tau_1 = 0.9178$ ns, $\tau_2 = 3.7443$ ns, $\tau_{ave} = 1.6641$ ns) increased by 5.8%, 19.8%, and 12.4%, respectively. As longer decay time (i.e. longer carrier lifetime) represents better film quality and fewer defects and traps, these demonstrate an overall reduction in traps, especially the deep trap and internal defects inside the film, which supports our proposed perspective.

Secondly, we supplemented quantitative experiments by using the space-charge-limited current (SCLC) method to quantify the trap density^{4,9,10}. We

fabricated vertical and hole-only devices (ITO-PEDOT-CNO-MoO₃-Ag), and measured their trap-filled limit voltages (V_{TFL}) to calculate the hole-trap densities (n_{trap}), as shown in Fig. R15. It is verified that the n_{trap} of S-CNO devices reduced more than 10 times compared with that of D-CNO device. Therefore, the reduction of traps and defects of S-CNO films compared with D-CNO film not only corresponds to the reduction of surface roughness (Supplementary Fig. 8h), but also supports our interpretation in Fig. 3g. More details in experiments and explanations are supplemented in our revised manuscript.

Fig. R15 | **a-e** Dark $I-V$ characteristics of hole-only devices measured using the space-charge-limited current (SCLC) method. **(a)** S-CNO 6 ml device, **(b)** S-CNO 8 ml device, **(c)** S-CNO 10 ml device, **(d)** S-CNO 12 ml device, and **(e)** D-CNO device. **f**, Statistics of the V_{TFL} and n_{trap} calculated from every three D-CNO and S-CNO devices.

In our revised manuscript, the corresponding changes are listed as follows:

① On the 786th line of page S11, the sentence “The table shows the fitted parameters of the TRPL spectrum. Note that the decay process consists of two components, including a fast decay process (τ_1) and a slow decay process (τ_2). While τ_1 is usually thought to be attributed to the surface recombination and the shallow-trap dominated trapping/de-trapping process, τ_2 is related to the internal recombination and the deep-trap dominated trapping/de-trapping process¹⁻³. Compared with the slow decay of D-CNO film ($\tau_1 = 0.8682$ ns, $\tau_2 = 3.1264$ ns, $\tau_{ave} = 1.4802$ ns), the τ_1 , τ_2 , and τ_{ave} of

S-CNO film ($\tau_1 = 0.9178$ ns, $\tau_2 = 3.7443$ ns, $\tau_{ave} = 1.6641$ ns) increased by 5.8%, 19.8%, and 12.4%, respectively. This demonstrates an overall reduction in traps, especially the deep trap and internal defects inside the film”.

② On the 251st line of page 12, the sentence “To provide a better illustration from a mechanistic point of view, we quantify the trap density (n_{trap}) by the space-charge-limited current (SCLC) method. As shown in Supplementary Fig. 19, the I - V curves of tailor-made hole-only S-CNO and D-CNO devices can be divided into three typical SCLC regions, and the n_{trap} of S-CNO devices that calculated from trap-filled limit voltages (V_{TFL}) is significantly lower than that of D-CNO device (Fig. 3f)”.

③ On the 459th line of page 23, the sentence “To fabricate the hole-only devices (ITO-PEDOT-CNO-MoO₃-Ag) for SCLC test, PEDOT solution was spin-coated on ITO substrate (3000 rpm for 30 s) and the annealed at 130 °C for 15 min, followed by the preparation of D-CNO and S-CNO films. Then, 10 nm MoO₃ was thermal evaporated on the films, followed with the evaporation of 30 nm Ag”.

④ On the 872nd line of page S15, the sentence “**(III) SCLC test:** As shown in **Supplementary Fig. 19**, the ohmic and trap-filled limit regions are distinctively separated at trap-filled limit voltage (V_{TFL}). The trap density (n_{trap}) can then be determined by the V_{TFL} with⁷⁻⁹

$$n_{trap} = \frac{2\varepsilon_0\varepsilon_r V_{TFL}}{eL^2}$$

where ε_0 denotes the vacuum permittivity of 8.85×10^{-12} F/m, ε_r is the dielectric constant of Ca₂Nb₃O₁₀ ($\varepsilon_r = 26$)¹⁰, e represents the elementary charge of 1.6×10^{-19} C, and L is the film thickness”.

⑤ On the 291st line of page 14, Fig. 3f has been added and Fig. 3 has been updated as shown below. Former Fig. 3f and Fig. 3g have been moved, former Fig. 3h has been moved to Supplementary Fig. 14, and corresponding captions have been updated.

⑥ On the 892nd line of page S17, **Supplementary Fig. 19** has been added.

Comment 2. *The surface of the mainstream two-dimensional material usually has van der Waals characteristics, with relatively low dangling bonds. The non-van der Waals characteristics of oxide perovskite nanosheets make them have high-density dangling bonds, which has an impact on the performance of the device? Does the residual solvent in the solution treatment process lead to the degradation of device performance?*

Response: Thank you for the valuable suggestion and comment and we will answer the two questions separately:

(1) First of all, we do understand your valuable concerns. It is well known that van der Waals 2D layered materials (such as *h*-BN and graphene) with chemically inert surfaces are free of dangling bonds, which are detrimental to device performance¹¹. And the surfaces of 2D non-layered materials are filled with dangling bonds¹². However, the Ca₂Nb₃O₁₀ nanosheets are synthesized via a special process, i.e. “soft chemical technique”, containing ion exchange, intercalation and exfoliation^{13,14}. In these processes, bulky organic ions like TBA⁺ weaken the ionic bonds between the negatively charged host layers and the cationic interlayer species into van der Waals between layers, and thereby facilitating peeling (as shown in Fig. S1)¹⁵. After exfoliation, superficial O atoms with lone electrons are too active to be stable and will grab protons from the aqueous solution to maintain stable surfaces. Moreover, latest research has claimed that “a unique characteristic of ionic crystals is that although

they do not possess vdW gap-like layered materials, their surface is still chemical inert and free of dangling bonds¹⁶”. And other recent studies about van der Waals integration between non-van der Waals oxide perovskite (such as $\text{Sr}_2\text{Nb}_3\text{O}_{10}$ ¹⁷ and SrTiO_3 ¹⁸) and MoS_2 have supported this opinion, as shown in Fig. R16. Therefore, there seems to be controversy about the dangling bonds on the surface of such non-van der Waals oxide perovskite nanosheets, which still needs to be further explored. We value and respect your constructive suggestion, and we will complement relevant discussion in the manuscript if you think it necessary.

Fig. R16 | Examples of dangling bonds-free integration at the interfaces between non-van der oxide perovskites and 2D layered van der Waals materials. **a**, Dangling bonds-free interface between oxide perovskite nanosheets $\text{Sr}_2\text{Nb}_3\text{O}_{10}$ and MoS_2 ¹⁷. **b** and **c**, van der Waals integration between 3D oxide perovskite SrTiO_3 and MoS_2 ^{18,19}.

(2) We understand your concern about the negative effect of residual solvent on device performance, which is a significant issue in almost all solution-processed devices. As shown in Fig. S6, there is much and obvious residual solvent in the dropped-coated and spin-coated films (Fig. S6b and c). In contrast, however, very little residual solvent can be observed in our spray-coated films (Fig. S6 d-i), which greatly eliminates the negative impact of solvent on the performance of our devices. We attribute this significantly reduced residual solvent to the rapid volatilization of the mixed solvent, which is caused by the addition of low-boiling anhydrous ethanol and the large-area nature of the spray-coating process. We have added corresponding discussion of this issue in the revised manuscript.

In our revised manuscript, the corresponding changes are listed as follows:

- ① On the 740th line of page S6, the sentence “**Note that it can be seen that there is much and obvious residual solvent in the dropped-coated (a and b) and spin-coated**

films (c). In contrast, very little residual solvent can be observed in our spray-coated films (d-i), which may greatly eliminate the negative impact of solvent on the device performance. The significantly fewer residual solvent to the rapid volatilization of the mixed solvent, which is caused by the addition of low-boiling anhydrous ethanol and the large-area nature of the spray-coating process”.

Comment 3. *Compared with the control group, the optimized device has higher recognition accuracy in the Motion recognition task. However, only comparing the devices prepared in their own two states cannot effectively reflect the advanced nature of the devices. It needs to be compared with commercial devices, such as silicon devices, GaAs, etc. for the same task.*

Response: We thank you for this comment and suggestion, and we do understand your concern.

First of all, comparing the two types of devices (experimental and control) mentioned in our work is more like that we are comparing our device with conventional devices, as all the 2D perovskite oxide-based devices presented in currently available studies are based on the dip-coating method used in the control group. That is, we tend to highlight the advanced nature of our device by comparing it with existing studies and exploring its prospects for application in the motion recognition area.

Secondly, since motion recognition is a relatively novel application in the field of photodetection, there are currently no commercially available devices. Meanwhile, as the program we have designed are based on our own devices, it is not yet possible to apply them in existing commercial devices to endow them with motion recognition capability. To the best of our knowledge, the newest related publications are also unable to make similar comparisons suggested²⁰⁻²³. However, it is believed that the device performance can to some extent reflect its application prospects in various fields, i.e. the better the overall performance, the more advantageous. Therefore, in response to your request, we have collected the performance of some commercial UV photodetectors and made comparisons to show the relative advancement of our

devices (Table R1). We hope this can address your concern and sincerely ask for your understanding. If you feel it is necessary, we will also put the following table into the revised manuscript.

Table R1. Comparisons between existing commercial UV photodetectors with our device

Material (Type number)	Wavelength (nm)	R _{max} (mA/W)	Rejection ratio R _{max} /R _{400 nm}	Dark current	Response speed	Company/Source
Ca ₂ Nb ₃ O ₁₀	240-330	11900 (280 nm)	185560	2×10 ⁻¹² (5 V)	42.4 μs / 1.77 ms	This work
Si (UV-100L)	200-1100	140 (254 nm)	NG	NG	5.9 μs	OSI Optoelectronics Inc. (osioptoelectronics.com)
Si (UV-100DQ)	190-1100	120 (200 nm)	NG	NG	3.0 μs	OSI Optoelectronics Inc. (osioptoelectronics.com)
Si (PIN-005D -254F)	230-280	25 (254 nm)	NG	NG	0.1 μs	OSI Optoelectronics Inc. (osioptoelectronics.com)
Si (MT03-023)	250-1100	220 (365 nm)	NG	2×10 ⁻¹⁰ (5 V)	100 ns	Marktech Optoelectronics Inc. (marktechopto.com)
GaN (GT-ABC-L)	210-370	200 (355 nm)	>10000	< 1×10 ⁻⁹ (-1 V)	NG	GaN Optoelectronics Inc. (gano-opto.com)
GaN (G365S01M)	220-370	180 (355 nm)	>1000	< 1×10 ⁻¹¹ (1 V)	NG	(uvdetek.com)
SiC (STL-ABC-XL)	220-350	900 (290 nm)	>10000	< 1×10 ⁻¹¹ (-5 V)	NG	GaN Optoelectronics Inc. (gano-opto.com)
SiC (MTD2800UV)	210-355	180 (265 nm)	NG	1×10 ⁻¹⁴ (1 V)	NG	Marktech Optoelectronics Inc. (marktechopto.com)
AlGaN (SMD 1010)	220-280	100 (NG)	NG	< 1×10 ⁻⁹ (NG)	NG	Genicom Co., Ltd. (genicom.en.ec21.com)
AlGaN (SD008-2171 -112)	220-280	60 (254 nm)	NG	1×10 ⁻¹² (0.1 V)	NG	Advanced Photonix (advancedphotonix.com)
NG (UVG20C)	300-1000	200 (400 nm)	NG	3×10 ⁻⁸ (13 V)	1.0 μs	Opto Diode Co. (optodiode.com)

NG: not given.

Comment 4. *Although the authors claim that the nanosheets have good orientation, the morphology of the nanosheets photographed by SEM is irregular (Fig.1c i-iii), and it seems that the nanosheets are stacked in the film. In this case, there is a large amount of lattice mismatch at the contact interface between the nanosheets. Will it cause charge capture to produce built-in electric field and other effects? For example,*

in Fig.3g, under certain illumination conditions, the voltage corresponding to the minimum current of the device is not 0, which means that there is a built-in electric field. Please discuss.

Response: We thank you for the question and advice. Our highly-oriented films are constructed based on the orderly stacking of nanosheets. The nanosheets in the revised Fig. 1b i-iii exhibit morphological irregularities, but this does not affect the orientation of the films (Fig. 2d-f). As you mentioned, we agree that the stacking of nanosheets does cause mismatch issue at the contact interfaces. And for the non-minimum current at 0 V in the revised Fig. 3h (original Fig. 3g), we suppose that there are three possible causes:

a. The most likely, as you suggest, is the built-in electric field due to the mismatch at the contact interface of nanosheets.

b. Secondly, the trapping and de-trapping process of carriers by traps and defects. In more detail, since this phenomenon occurs only in dark and at low light densities (as shown in Fig. R17), the number of photoinduced carriers is relatively small in these cases. It will take some time for these fewer carriers to be trapped and de-trapped by the defects inside the film, which results in a hysteresis in the change of photocurrent, i.e., the change of current cannot keep up with the change of voltage (scanning from -5 V to 5 V) during the I - V test.

c. Thirdly, the systematic error caused by test instruments and operations, such as subtle contact differences between the probe tips of the semiconductor test system on Au electrodes, may also lead to such small differences. This is relatively very common in most of the literature, and we believe that this minor deviation imposes little impact on the presentation of our work.

We have supplemented related discussions in our revised manuscript.

Fig. R17 | Photoresponse of the S-CNO device in dark and under 280 nm illumination with different light intensities.

In our revised manuscript, the corresponding changes are listed as follows:

① On the 278th line of page 13, the sentence “The I - V curves under 280 nm UV illumination with different light intensities in ~~Fig. 3g~~ Fig. 3h show that the photosensitivity gradually increases as the light intensity increases. **Incidentally, we suppose that the non-minimum currents at 0 V bias occurring in dark and at low light densities may originate from the mismatch-induced built-in electric field at the interfaces of nanosheets and the relatively slower trapping and de-trapping process of fewer carriers by defects and traps in these cases**”.

Comment 5. *Some pictures need to be corrected or further discussed.*

a. Fig 3b needs to add errorbar.

b. Fig.3i suggests increasing the comparison of halide perovskite and other two-dimensional material categories.

c. Fig 3g shows why the voltage value corresponding to the minimum current value under different light conditions is different (e.g., from bottom to top line 3-5).

Response: We appreciate your constructive suggestions to make us refine our work.

a. As shown in Fig. R18a, according to your suggestion, we have re-collected relevant data and added the error bars in Fig. 3b.

b. As shown in Fig. R18b, according to your suggestion, we have complemented Fig. 3i with the comparisons of representative halide perovskites and other two-dimensional materials that used for UV detection concluded from Supplementary Table1 and Table 2.

c. From our point of view, this is the same issue as that raised in Comment 4, and we

have explained in detail in our response to Comment 4.

Fig. R18 | **a**, Modified photocurrents and on-off ratios of S-CNO devices measured under 5 V bias and 280 nm UV illumination (703 μW cm⁻²) **b**, Modified performance comparison of representative 2D perovskite oxide-, 2D halide perovskite- and other 2D material-based film UV photodetectors, in which the data are concluded from Supplementary Table 2.

In our revised manuscript, the corresponding changes are listed as follows:

① On the 291st line of page 14, Fig. 3b has been updated as show below.

② On the 291st line of page 14, Fig. 3i has been updated as shown below and related statement has been added to its caption: “Performance comparison of representative 2D perovskite oxide-, 2D halide perovskite- and other 2D material-based film UV photodetectors, in which the data are concluded from Supplementary Table 2”.

③ The corresponding interpretations as the response to Comment 4 has been supplemented in our revised manuscript.

References

1. ten Elshof, J. E., Yuan, H. & Gonzalez Rodriguez, P. Two-dimensional metal oxide and metal hydroxide nanosheets: synthesis, controlled assembly and applications in energy conversion and storage. *Adv. Energy Mater.* **6**, 1600355 (2016).
2. Li, Z., Hong, E., Zhang, X., Deng, M. & Fang, X. Perovskite-type 2D materials for high-performance photodetectors. *J. Phys. Chem. Lett.* **13**, 1215-1225 (2022).
3. Abdelsamie, M. et al. Impact of processing on structural and compositional evolution in mixed metal halide perovskites during film formation. *Adv. Funct. Mater.* **30**, 2001752 (2020).
4. Chen, J. et al. Single-crystal thin films of cesium lead bromide perovskite epitaxially grown on metal oxide perovskite (SrTiO₃). *J. Am. Chem. Soc.* **139**, 13525-13532 (2017).
5. Guo, T. et al. Large-area metal-semiconductor heterojunctions realized via MXene-induced two-dimensional surface polarization. *ACS Nano* **17**, 8324-8332 (2023).
6. Liu, X. et al. Enhanced response speed in 2D perovskite oxides-based photodetectors for UV imaging through surface/interface carrier-transport modulation. *ACS Appl. Mater. Interfaces* **14**, 48936-48947 (2022).
7. Mo, X. et al. Highly-efficient all-inorganic lead-free 1D CsCu₂I₃ single crystal for white-light emitting diodes and UV photodetection. *Nano Energy* **81**, 105570 (2021).
8. Wen, Z. et al. Graphene induced structure and doping level tuning of evaporated CsPbBr₃ on different substrates. *Chem. Eng. J.* **452**, 139243 (2023).
9. Dong, S. et al. All-inorganic perovskite single-crystal photoelectric anisotropy. *Adv. Mater.* **34**, 2204342 (2022).
10. Liu, Y. et al. A 1300 mm² ultrahigh-performance digital imaging assembly using high-quality perovskite single crystals. *Adv. Mater.* **30**, 1707314 (2018).
11. Illarionov, Y. Y. et al. Insulators for 2D nanoelectronics: the gap to bridge. *Nat.*

- Commun.* **11**, 3385 (2020).
12. Wang, F. et al. Two-dimensional non-layered materials: Synthesis, properties and applications. *Adv. Funct. Mater.* **27**, 1603254 (2016).
 13. Kweon, S.-H. et al. Electrophoretic deposition of $\text{Ca}_2\text{Nb}_3\text{O}_{10}^-$ nanosheets synthesized by soft-chemical exfoliation. *J. Mater. Chem. C* **4**, 178-184 (2016).
 14. Xu, F., Ebina, Y., Bando, Y. & Sasaki, T. Structural characterization of (TBA, H) $\text{Ca}_2\text{Nb}_3\text{O}_{10}$ nanosheets formed by delamination of a precursor-layered perovskite. *J. Phys. Chem. B* **107**, 9638-9645 (2003).
 15. Ma, R. & Sasaki, T. Nanosheets of Oxides and Hydroxides: Ultimate 2D Charge - Bearing Functional Crystallites. *Adv. Mater.* **22**, 5082-5104 (2010).
 16. Yang, S. et al. Gate dielectrics integration for 2D electronics: Challenges, advances, and outlook. *Adv. Mater.* **35**, 2207901 (2023).
 17. Li, S., Liu, X., Yang, H., Zhu, H. & Fang, X. Two-dimensional perovskite oxide as a photoactive high- κ gate dielectric. *Nat. Electron.* **7**, 216-224 (2024).
 18. Yang, A. J. et al. Van der Waals integration of high- κ perovskite oxides and two-dimensional semiconductors. *Nat. Electron.* **5**, 233-240 (2022).
 19. Zhou, W., Zhang, S. & Zeng, H. Perovskite oxides as a 2D dielectric. *Nat. Electron.* **5**, 199-200 (2022).
 20. Chen, J. et al. Optoelectronic graded neurons for bioinspired in-sensor motion perception. *Nat. Nanotechnol.* **18**, 882-888 (2023).
 21. Cho, H. et al. Real-time finger motion recognition using skin-conformable electronics. *Nat. Electron.* **6**, 619-629 (2023).
 22. Sun, L. et al. Bio-Inspired Vision and Neuromorphic Image Processing Using Printable Metal Oxide Photonic Synapses. *ACS Photonics* **10**, 242-252 (2022).
 23. Zhan, Z. et al. A perovskite photodetector crossbar array by vapor deposition for dynamic imaging. *Adv. Mater.* **34**, 2207106 (2022).

In our revised manuscript, the revisions are listed below in order:

1. On the 91st line of page 5, the sentence “Given the special intrinsic negative charge of CNO (Supplementary Fig. 2), to achieve the large-area homogeneous preparation, we ~~further~~ introduce a charge-assisted oriented assembly film-formation (COAF) strategy (~~see Methods section for more details~~)”.
2. On the 100th line of page 5, the sentence “The solvent mixture of deionized water and ethanol not only provides reduced surface ~~tension~~ energy for better dispersion of solutes ~~and enhanced wettability of the solution on substrates (Supplementary Fig. 4)~~, but also enables the nanosheets to instant stack layer-by-layer with the rapid volatilization of solvent to avoid the percolation issue”.
3. On the 104th line of page 5, the sentence “As shown in Fig. 1a, in this process, ~~the sprayed large number of micro-droplets first ensure the large-area in-plane spreading on the substrate (Fig. 1a, i). And~~ the repulsive forces induced by the intrinsic negative charges between the nanosheets in micro-droplets prevent aggregation and facilitate their dispersion (Fig. 1a, ~~i and~~ ii)”.
4. On the 112nd line of page 6, the sentence “As the solvent evaporates, the nanosheets ~~in each micro-droplet~~ settle and stack layer by layer, ~~accompanied by the filling of voids and gaps between the larger nanosheets by the tinier ones (Supplementary Fig. 5). But given the high shape irregularity of the nanosheets, the formed films may still not be totally void-free. After the complete evaporation of solvent, forming~~ tight structures with high degree of orientation are formed (Fig. 1a, v). ~~Meanwhile, since the large number of micro-droplets cover the substrate surface uniformly over a large area during the spraying process, the nanosheets within each droplet will connect with those adjacent ones to form an integral whole. Ultimately, the ordered stack and the connections covering the whole substrate of the nanosheets simultaneously completes the construction of the highly oriented film (Fig. 1a, vi)~~”.
5. On the 137th line of page 7, the sentence “Quantitative roughness characterized by AFM also reveals that the surfaces of the S-CNO films are significantly smoother

- (~~Fig. 1d and~~ Supplementary Fig. 8)”.
6. On the 139th line of page 7, the sentence “Incidentally, ~~these images still show some relatively large particles which are believed to be thicker nanosheets and incompletely exfoliated HCa₂Nb₃O₁₀ (HCNO), and better optimization of film structure is expected to be realized by optimizing the synthesis...~~”.
7. On the 145th line of page 7, ~~Fig. 1a (v)~~ has been added and ~~Fig. 1~~ has been updated. ~~Fig. 1d~~ has been moved to Supplementary Fig. 8.
8. On the 146th line of page 7, the sentence “~~a, Schematic of the Charge-assisted oriented assembly film-formation (COAF) process of the S-CNO film (i to vi). Specific interpretations are presented in Supplementary Note I~~”.
9. On the 158th line of page 8, the sentence “...illustrating that the nanosheets are all aligned along [001], as shown in Fig. 2a. ~~Note that the leftward shifts of the diffraction peaks in the low-angle region originate from the insertion of large TBA⁺ cations, as the increased interplanar spacing leads to a decrease in the diffraction angles.~~—Accordingly, we plot the micro-schematic of the cross-section...”.
10. On the 223rd line of page 11, the sentence “~~With a channel area of 50000 μm², as shown in Fig. 3c, the optimal S-CNO 10 ml photodetector achieves a prime responsivity and a remarkable detectivity of 11.9 A/W and 3.71 × 10¹⁴ Jones under 280 nm UV illumination and 5 V bias~~”.
11. On the 251st line of page 12, the sentence “To provide a better illustration from a mechanistic point of view, ~~we quantify the trap density (n_{trap}) by the space-charge-limited current (SCLC) method. As shown in Supplementary Fig. 19, the I - V curves of tailor-made hole-only S-CNO and D-CNO devices (ITO-PEDOT-CNO-MoO₃-Ag) can be divided into three typical SCLC regions, and the n_{trap} of S-CNO devices that calculated from trap-filled limit voltages (V_{TFL}) is significantly lower than that of D-CNO device (Fig. 3f)~~”.
12. On the 275th line of page 13, the sentence “~~Experimental results and interpretations in Supplementary Fig. 13 also support this perspective~~”.
13. On the 278th line of page 13, the sentence “The I - V curves under 280 nm UV

illumination with different light intensities in ~~Fig. 3g~~ Fig. 3h show that the photosensitivity gradually increases as the light intensity increases. Incidentally, we suppose that the non-minimum currents at 0 V bias occurring in dark and at low light densities may originate from the mismatch-induced built-in electric field at the interfaces of nanosheets and the relatively slower trapping and de-trapping process of fewer carriers by defects and traps in these cases”.

14. On the 287th line of page 13, the sentence “The fitted nonunity indexes of the law ($\theta > 1$) ~~in Supplementary Fig. 14~~ are related to the complicated processes of generation...”.
15. On the 291st line of page 14, Fig. 3b has been updated.
16. On the 291st line of page 14, Fig. 3f has been added and related statement has been added to its caption. Former Fig. 3f and Fig. 3g have been moved, former Fig. 3h has been moved to Supplementary Fig. 14.
17. On the 291st line of page 14, Fig. 3i has been updated and related statement has been added to its caption: “Performance comparison of representative 2D perovskite oxide-, 2D halide perovskite- and other 2D material-based film UV photodetectors, in which the data are concluded from Supplementary Table 2”.
18. On the 293rd line of page 14, the sentence “Photocurrents and on-off ratios of S-CNO devices measured under ~~±~~ 5 V bias and 280 nm UV illumination”.
19. On the 301st line of page 15, the sentence “...and the bottom diagrams show the practical photo-switching response”.
20. On the 305th line of page 15, the sentence “Note that the performance data in Fig. 3 and corresponding context are measure on D-CNO and S-CNO devices with channel areas of 50000 μm^2 ”.
21. On the 319th line of page 15, the sentence “The overall structure of the active area is schematically shown in Fig. 4a, and the inset shows a channel area of 4500 μm^2 for each individual device”.
22. On the 326th line of page 15, the sentence “...with a low variation coefficient (C_V) of 17.6% under 280 nm illumination and 1 V bias, as shown in Fig. 4d and Supplementary Fig. 24. Note that the relatively low photocurrents stem from the

reduction in device channel and applied bias and this does not mean the significant performance decline compared with rigid device (Supplementary Fig. 24)”.

23. On the 335th line of page 16, the sentence “Subsequently, the arrays are bent to 90°, and the *I-t* curves of fifty pixels in the bending state are randomly measured. ~~to display the functionality of the array in flexible scenarios (Fig. 4f).~~ As shown in Fig. 4f, the response behaviors of these pixels are essentially identical, and their photocurrents remain highly uniform with a C_v of 17.3%. It can be seen that morphology transformations do not interfere with the performance consistency, and consequently the device arrays in the bending state still show decent imaging result (Fig. 4g). Moreover, ultra-flexibility also allows the device array to conform to substrates of arbitrary shapes. By transferring the array onto a glass hemispherical substrate, a UV-sensitive retina-like device is presented (Fig. 4h), which still also exhibits clear and stable imaging capability (Fig. 4i)”.
24. On the 347th line of page 17, the sentence “The inset shows an optical microscopy image of an individual S-CNO device with a pair of interdigital electrodes and a channel area of 4500 μm^2 ”.
25. On the 353rd line of page 17 (regarding Fig. 4f), the sentence “*I-t* characteristics of 50 pixel-units in the 90° bending state. g, Bending imaging results under 90°”.
26. On the 353rd line of page 18, the sentence “Photograph of the flexible array conformed to a hemispherical glass mold. Scale bar: 1 cm”.
27. On the 371st line of page 18, the sentence “As the schematic processing flow in Fig. 5a shows, when the target object moves (take a falling straw hat as an example), the pixels in the sensor that receive the light...”
28. On the 403rd line of page 20, the sentence “Schematic illustration of spatiotemporal vision imaging acquisition and motion recognition realization”.
29. On the 459th line of page 23, the sentence “To fabricate the hole-only devices (ITO-PEDOT-CNO-MoO₃-Ag) for SCLC test, PEDOT solution was spin-coated on ITO substrate (3000 rpm for 30 s) and the annealed at 130 °C for 15 min, followed by the preparation of D-CNO and S-CNO films. Then, 10 nm MoO₃ was

- thermal evaporated on the films, followed with the evaporation of 30 nm Ag”.
30. On the 487th line of page 24, the sentence “The contact angles were measured using a contact angle analyzer (OCA25, Dataphysics, Germany), and the average values were adopted from at least three parallel measurements using a 2 μ L droplet”.
 31. On the 489th line of page 24, the sentence “The flexible tests were performed on the flexible electronic tester (FT2000)”.
 32. On the 490th line of page 24, the sentence “Note that all error bars in the bar charts and dot plots in the manuscript represent the standard error, and the box plots contain the median line, mean value, outlier that lies beyond the whiskers, box limit from the lower quartile to the upper quartile, and whiskers that extend from the box to the smallest and largest values within 1.5 times the IQR from the first and third quartiles”.
 33. On the 523rd line of page 25, the sentence “Code availability. The codes and dataset that used for designing and training the convolutional neural network are available from the corresponding authors upon reasonable request”.
 34. On the 529th line of page 26, the sentence “Thanks to Zijun Hu for performing the SEM characterizations and taking photos in Fig. 4. Thanks to Tingting Yan for performing the AFM characterizations. Thanks to Zijin Zhao for helping with the GIWAXS test”.
 35. On the 607th line of page 28, reference 29 has been added.
 36. On the 665th line of page S3, related discussions have been added to the caption of Supplementary Fig. 1: “Note that after acid converts $\text{KCa}_2\text{Nb}_3\text{O}_{10}$ to $\text{HCa}_2\text{Nb}_3\text{O}_{10}$, the protons in $\text{HCa}_2\text{Nb}_3\text{O}_{10}$ are then replaced by TBAOH treatment to achieve exfoliation, and finally TBA^+ cations are washed away to produce $\text{Ca}_2\text{Nb}_3\text{O}_{10}^-$ nanosheets. In the above process, negative charges are introduced into the nanosheets due to the stripping of interlayer cations, which is proved in **Supplementary Fig. 2**”.
 37. On the 671st line of page S3, **Supplementary Fig. 2** has been added and related discussions have been added to its caption: “After several months of placement,

the nanosheets remains dispersed in solution without obvious settling or aggregation, proving the existence of repulsive forces between the nanosheets”.

38. On the 682nd line of page S4, **Supplementary Fig. 4** has been added and related discussions have been added to its caption: “As the concentration of ethanol increases, the contact angles of the solvent mixture droplets on both glass and Parylene-C decrease, suggesting that the increase in ethanol reduces the surface energy of the solvent and improves its wettability with the substrates”.
39. On the 696th line of page S4, **Supplementary Note I** has been supplemented and updated: “For processes i-~~v~~ vi in Fig. 1a: (i) The large number of tiny micro-droplets ejected from the gun ensure a uniform distribution of nanosheets, and as they cover the surface of the substrate, they also ensure the large-area in-plane spreading parallel to the substrate”.
40. On the 699th line of page S5, the sentence “(ii) For each individual micro-droplet, as the top-down exfoliation endows CNO nanosheets with intrinsic negative charges, the resultant repulsive forces prevent the aggregation of nanosheets, further contributing to their dispersion inside the droplets”.
41. On the 707th line of page S5, the sentence “(iv) Since the prepared nanosheets precursor solution contains nanosheets with lateral size ranging from a few micrometers (larger ones) to tens of nanometers (tinier ones), it is also necessary to consider the behavior and significant role of tinier nanosheets in the COAF process – as the solvent evaporates and penetrates, they will ~~smaller nanosheets~~ fill in the gaps and voids to ensure the integrity of the films ~~as the solvent penetrates~~ (as details shown in **Supplementary Fig. 5**). Note that there are two main forces of the void-filing process: a. the same repulsion between the nanosheets (preventing tinier nanosheets from settling directly on the surface of larger nanosheets), b. the percolation effect of the solvent accompanied by volatilization. Meanwhile, the larger nanosheets gradually settle in this ordered arrangement as the solvent ~~dissipates volatilizes under heating~~. However, considering the irregular shape of the nanosheets, there may still be some inevitable voids inside the nanosheets films”.

42. On the 718th line of page S5, the sentence “(v) After the solvent completely evaporates, the layer-by-layer stackings of nanosheets ~~form films with~~ in each individual micro-droplet ensure high degrees of orientation and tight structures, ~~and uniform surfaces~~. At the same time, due to the large-area and uniform coverage of the substrate surface by a large number of micro-droplets during the spray-coating process, the nanosheets inside the droplets will connect with the adjacent ones to form a block of thin film while settling, forming the large-area in-plane spreading of nanosheets parallel to the substrate surface. (vi) Ultimately, the ordered stack and the connections covering the entire substrate of the nanosheets simultaneously completes the construction of the highly oriented film”.
43. On the 727th line of page S6, **Supplementary Fig. 5** has been added and related interpretations have been added to its caption.
44. On the 740th line of page S6, the sentence “Note that it can be seen that there is much and obvious residual solvent in the dropped-coated (a and b) and spin-coated films (c). In contrast, very little residual solvent can be observed in our spray-coated films (d-i), which may greatly eliminate the negative impact of solvent on the device performance. The significantly fewer residual solvent to the rapid volatilization of the mixed solvent, which is caused by the addition of low-boiling anhydrous ethanol and the large-area nature of the spray-coating process”.
45. On the 749th line of page S7, the sentence “Note that large size nanosheets are observed on the surfaces of both D-CNO and S-CNO films, which are formed by incomplete exfoliation. And the nanosheets on the surface of D-CNO seem to be more and larger, which we believe is due to: (1) Difference in the film-formation process. Large nanosheets in S-CNO films tend to be buried inside and form uniform surfaces, while that in D-CNO films are exposed on the surfaces. (2) As relatively more precursor solution per unit area is needed to prepare the D-CNO film, there will be more large nanosheets in them than in S-CNO films, while the large nanosheets on the S-CNO films appear to be more dispersed”.

46. On the 776th line of page S10, **Fig. S12b** has been updated.
47. On the 777th line of page S10, the sentence “The transmittance of S-CNO films with different thicknesses **tested under 280 nm UV**”.
48. On the 786th line of page S11, the sentence “The table shows the fitted parameters of the TRPL spectrum. **Note that the decay process consists of two components, including a fast decay process (τ_1) and a slow decay process (τ_2). While τ_1 is usually thought to be attributed to the surface recombination and the shallow-trap dominated trapping/de-trapping process, τ_2 is related to the internal recombination and the deep-trap dominated trapping/de-trapping process¹⁻³. Compared with the slow decay of D-CNO film ($\tau_1 = 0.8682$ ns, $\tau_2 = 3.1264$ ns, $\tau_{ave} = 1.4802$ ns), the τ_1 , τ_2 , and τ_{ave} of S-CNO film ($\tau_1 = 0.9178$ ns, $\tau_2 = 3.7443$ ns, $\tau_{ave} = 1.6641$ ns) increased by 5.8%, 19.8%, and 12.4%, respectively. This demonstrates an overall reduction in traps, especially the significant reduction in deep trap and internal defects inside the film”.**
49. On the 872nd line of page S15, the sentence “**(III) SCLC test: As shown in **Supplementary Fig. 19**, the ohmic and trap-filled limit regions are distinctively separated at trap-filled limit voltage (V_{TFL}). The trap density (n_{trap}) can then be determined by the V_{TFL} with⁷⁻⁹**

$$n_{trap} = \frac{2\varepsilon_0\varepsilon_r V_{TFL}}{eL^2}$$

where ε_0 denotes the vacuum permittivity of 8.85×10^{-12} F/m, ε_r is the dielectric constant of $\text{Ca}_2\text{Nb}_3\text{O}_{10}$ ($\varepsilon_r = 26$)¹⁰, e represents the elementary charge of 1.6×10^{-19} C, and L is the film thickness”.

50. On the 892nd line of page S17, **Supplementary Fig. 19** has been added.
51. On the 988th line of page S22, **Supplementary Fig. 24** has been updated and related discussions have been added to its caption: “**Note that, while exhibiting an average photocurrent of $(9.08 \pm 1.60) \times 10^{-9}$ A and an average dark current of $(5.24 \pm 6.28) \times 10^{-12}$ A, these flexible devices reveal an average responsivity of 0.29 A/W under 1 V bias and 280 nm illumination, which is very close to the that of rigid device (0.85 A/W) under the same test conditions**”.

52. On the 996th line of page S23, **Supplementary Fig. 25** has been supplemented and related discussions have been added to its caption: “**It can be seen that significant performance degradation of the device occurs when the bending angle is larger than 150°**”.
53. **Fig. 1a, Fig. 3b, Fig. 3f and Fig. 3i** have been updated.
54. **Supplementary Fig. 2, Supplementary Fig. 4, Supplementary Fig. 5, and Supplementary Fig. 19** have been added to the Supplementary Information.
55. **Supplementary Fig. 8, Supplementary Fig. 12, Supplementary Fig. 17, Supplementary Fig. 24, and Supplementary Fig. 25** have been updated.
56. All **figure numbers** have been updated.

REVIEWERS' COMMENTS

Reviewer #1 (Remarks to the Author):

In this work, the authors apply the constructed 16*16 device array for the motion recognition part with the help of convolutional neural network design and training. I believe that this part can be seen as complementing and justifying the innovative design of the material assembly, while demonstrating the applicability of the device in motion recognition applications.

Regarding the concerns on this part raised by Reviewer #2 (mainly Comment 13), I think the authors' responses are reasonable. First, they illustrate the mechanism of how the motion recognition function is achieved by the 256-pixel device array in Fig.5 and have shown the differences between the control and experimental groups. Secondly, I agree that the innovation of the motion recognition part of this manuscript lies more in the mechanism.

Although additional video may enhance the presentation of experimental results, empirically, this presentation effect also depends on other factors such as the number of array pixels.

This would require further up-scaling of the device structure and corresponding adjustments to the neural network, which is an engineering improvement instead of fundamental advance. Thirdly, the authors mentioned that the dataset they collected based on the array includes motions in eight directions. Since the final accuracy (Fig. 5e) is obtained from multiple directions, the feasibility of "multiple distinct motion" has been verified.

As for the comments raised by Reviewer #3 (mainly Comment 3), the authors' responses are also rational. As they say in the response, because of the proposed material assembly strategy, it is reasonable to say that their comparison is based on the comparisons of conventional devices, not just "their own two states". Due to the variability in the structures of general commercially available devices and their compatibility with specific neuromorphic networks, it is difficult to "reflect the advanced nature of devices" by such application-specific comparisons. At the same time, the authors' response sufficiently compares the performance of their device with that of many existing commercial devices, which are convincing and can illustrate the issue.

Overall, the authors have provided adequate responses to the concerns and comments raised by Reviewers #2 and #3, which are reasonable, and the general additional experiments and interpretations are also convincing. Therefore, from my side, it is recommended this revised manuscript to be published in Nature Communications as it is.

Reviewer #2 (Remarks to the Author):

The authors put significant effort into addressing the comments. However, despite their efforts, the comments have not been properly addressed. I was expecting the authors can fully address my comments in the revision, but unfortunately I cannot conclude the authors have thoroughly reflected the comments. Therefore, in my opinion, this manuscript does not seem suitable for publication in Nature Communications. The reasons are detailed below.

Firstly, the explanation on the mechanism behind the fabrication of 2D perovskite oxide nanosheets films is not entirely convincing. As the authors stated that "the prepared nanosheets solution contains nanosheets with lateral size ranging from a few micrometers (larger ones) to tens of nanometers (tinier ones)" and that "We believe that it is the tinier nanosheets that effectively filled the voids (between larger nanosheets)." This explanation implies that still tinier voids would remain between the smaller nanosheets at the nanometer scale, which could be a critical defect affecting charge transport in 2D semiconductor devices. Additionally, the scheme in Fig. 1a does not fully address the issues in fabrication

process.

Furthermore, although Fig. 3i suggests that this work represents a state-of-the-art advancement among the 2D semiconductor photodetectors, it should be compared with other types of dimensional semiconducting devices, not just 2D materials, which might make the work seem to be the world record but actually it isn't. This broader comparison could reveal whether the work reports the genuinely meaningful figure of merits in photodetector performance.

Lastly, to my knowledge, the main objective of the paper appears to be high-performance motion recognition, but it does not offer significant advancements over existing photodetector arrays. That's why I recommended to provide complementary video which would be an important data for comparison with recognition data based on the neuromorphic network.

Reviewer #4 (Remarks to the Author):

I thoroughly examined the revised manuscript, the response letter, and the final evaluation provided by the previous reviewers. I believe that this study has contained various sub-topics that the author has emphasized as novel compared to previous studies. The reviewer's assessment of this study varied significantly depending on which points were deemed essential and primary. Nevertheless, I would prefer to assign greater importance to the final assessment provided by reviewer #1 due to the following reasons.

First, as the revised manuscript noted, the author stated only the role of their device as photosensor that can output a fine single-frame imaging for the motion recognition. The recognition capability can be greatly influenced by various factors such as the choice of algorithm, the size and number of kernels used in the signal processing of the CNN, and the datasets utilized, which can not be only determined by the performance of the photosensor. In my opinion, the primary innovation of this study, which I discovered, remains valuable irrespective of whether additional video demonstrations of motion recognition are included. Also, if necessary, the author already responded that they are willing to an additional video (page 32 of the response letter). Plus, I don't think that the main and key objective of this paper is high-performance motion recognition. I believe that this was demonstrated as an example of the electronic applications of wafer-scale 2D perovskite oxide, specifically focusing on the function of the photosensor.

Second, as this study argues and offers a significant and strategic approach for the efficient integration of 2D perovskite oxide semiconductor and its synthesis on a large scale as the main results (I agree these are main findings in this study), it is reasonable to compare the performance of this material with other similar groups of 2D materials, such as 2D perovskite oxide and 2D halide perovskite, rather than solely focusing on the photosensor's performance of other material groups (Figure 3i in the revised manuscript). In this comparison, this suggested method and its performance looks superior.

Third, regarding the production of 2D perovskite oxide nanosheet films and its mechanism, I believe that they have adequately elucidated the subject and conducted supplementary experiments why this method could be utilized as alternative way for uniform and wafer-scale synthesis beyond other methods. Furthermore, they sufficiently discussed and contained the remaining challenges and potential constraints of their proposed method regarding the performance of the device in the revised manuscript and their response letter.

Thus, I concur with reviewer #1's assessment that this revised manuscript has effectively resolved all the concerns raised by the reviewers, and I agree that it is suitable for publication in Nature Communications.

Response to the reviewer's comments

Reviewer #2

Comment 1. *Firstly, the explanation on the mechanism behind the fabrication of 2D perovskite oxide nanosheets films is not entirely convincing. As the authors stated that “the prepared nanosheets solution contains nanosheets with lateral size ranging from a few micrometers (larger ones) to tens of nanometers (tinier ones)” and that “We believe that it is the tinier nanosheets that effectively filled the voids (between larger nanosheets).” This explanation implies that still tinier voids would remain between the smaller nanosheets at the nanometer scale, which could be a critical defect affecting charge transport in 2D semiconductor devices. Additionally, the scheme in Fig. 1a does not fully address the issues in fabrication process.*

Response:

We appreciate you for the careful comment. Regarding your concern on the issue of explanation of the film-forming mechanism, first, we believe that with your constructive comments, we have formed a rational and comprehensive explanation of the film-formation mechanism by explaining more details like the void-filling process and complementing the in-plane mechanism in our previously revised manuscript. We greatly appreciate your kind suggestion and contribution.

Second, as for the voids you mentioned between 2D nanosheets, we think that they are indeed inevitable and common at the nanometer scale after these nanosheets form a film^{1,2}. This is determined by the 2D morphology of the nanosheet materials themselves, unless these voids can be eliminated by measures such as the addition of additives³. And this issue is attributed to the limitation caused by the current synthesis process of perovskite oxide-type nanosheets, and to ‘fully address the issues in fabrication process’ is believed to depend on the further complete improvement of the synthesis method to achieve large-area thin film of single-crystalline perovskite oxides in the future.

Third, in this context, the advantage of our work lies in the optimization of the oriented alignment and stacking of the nanosheets through unique charge forces thereby minimizing the presence of voids, especially the voids between the nanosheet

layers, while ensuring the feasibility of large-area scale fabrication, as shown in Fig. R1a. And actually, we believe that these tinier voids can be further filled with much tinier nanosheets (below tens of nanometers) so that the intact films shown in Fig. R1b can be practically obtained. Furthermore, in this process, we have substantially reduced the number of voids and defects inside the film compared to the conventional method (as shown in Fig. R1c). Therefore, we believe this is a major innovation for the integrated fabrication of 2D nanosheets with varied sizes, and we have also verified this from both theoretical analysis and experimental results perspectives.

Fig. R1 | **a**, Schematic of the Charge-assisted oriented assembly film-formation (COAF) process of the S-CNO film. **b**, Surface morphology SEM images of the S-CNO film during the COAF process. Scale bar: 1 μm . **c**, Statistics of the V_{TFL} and n_{trap} calculated from every three hole-only D-CNO and S-CNO devices.

Comment 2. Furthermore, although Fig. 3i suggests that this work represents a state-of-the-art advancement among the 2D semiconductor photodetectors, it should be compared with other types of dimensional semiconducting devices, not just 2D materials, which might make the work seem to be the world record but actually it isn't. This broader comparison could reveal whether the work reports the genuinely meaningful figure of merits in photodetector performance.

Response:

We thank you for the comment. First, the significant objects and focus of our work are on the nanosheets with unique 2D structures, therefore, we chose the newest and representative materials (i.e., novel 2D perovskite oxides, and 2D halide perovskite based on Reviewer #3's suggestion) to make the unbiased comparisons in the most relevant areas. Second, indeed, although comparisons 'with other types of dimensional semiconducting devices' are feasible, excessive but less relevant comparisons may detract from the key point we are trying to highlight and weaken the emphasis and true innovation. Also, just as Reviewer #4 states, since we focus more on the meaningful approach for the efficient integration of 2D perovskite oxides and their fabrication on a large scale, we believe that comparing with other similar groups of 2D materials seems to be adequate and more reasonable, and focusing more on the comparisons of photodetector's performance with other dimensional semiconductors seems to be relatively less beneficial to our target.

Comment 3. Lastly, to my knowledge, the main objective of the paper appears to be high-performance motion recognition, but it does not offer significant advancements over existing photodetector arrays. That's why I recommended to provide complement supplementary video which would be an important data for comparison with recognition data based on the neuromorphic network.

Response:

We thank you for the comment and we totally understand your concern that providing a supplementary video helps us complete and refine our work. However, first, as we stated in our response letter to your Comment 13 (page 32), since the schematic illustrations Fig. 5a, Fig. 5c, and Fig. 5d have already shown our whole processing flow, our pre-preparation and attempts proved that the video is not a significantly improvement over these illustrations. Moreover, as Reviewer #1 said, it seems that the supplementary video may not contribute to our innovations in terms of mechanism and principle too much, and may even focus more on the relatively less important representation.

Second, as stated by Reviewer #4, the realization of high-performance motion

recognition relies on various and multiple factors. Therefore, comparisons with existing photodetector arrays require simultaneous consideration of factors like neural network architecture, array pixels, datasets, and other elements, which to the best of our knowledge are quite different from one work to another⁴⁻⁷. Meanwhile, the more important part of our work lies in the materials fabrication, especially the special wafer-scale integration process, and motion recognition is a potential and effective application direction based on this primary innovation. In conclusion, these are the reasons why we did not provide a supplementary video.

References

1. Xue, J. et al. Solution-processable assembly of 2D semiconductor thin films and superlattices with photoluminescent monolayer inks. *Chem* **10**, 1471-1484 (2024).
2. Huang, L., Wu, H., Ding, L., Caro, J. & Wang, H. Shearing liquid-crystalline MXene into lamellar membranes with super-aligned nanochannels for ion sieving. *Angew. Chem. Int. Ed.* **63**, 202314638 (2024).
3. Li, W. et al. Ultrastrong MXene film induced by sequential bridging with liquid metal. *Science* **385**, 62-68 (2024).
4. Chen, J. et al. Optoelectronic graded neurons for bioinspired in-sensor motion perception. *Nat. Nanotechnol.* **18**, 882-888 (2023).
5. Liao, F. et al. Bioinspired in-sensor visual adaptation for accurate perception. *Nat. Electron.* **5**, 84-91 (2022).
6. Zhan, Z. et al. A perovskite photodetector crossbar array by vapor deposition for dynamic imaging. *Adv. Mater.* **34**, 2207106 (2022).
7. Luo, X. et al. A bionic self-driven retinomorphing eye with ionogel photosynaptic retina. *Nat. Commun.* **15**, 3086 (2024).

In our revised manuscript, the revisions are listed below in order:

1. On the 7th line of page 1, the authors' affiliation 'Shanghai Frontiers Science Research Base of Intelligent Optoelectronics and Perception, Institute of Optoelectronics, **State Key Laboratory of Photovoltaic Science and Technology**, Fudan University, Shanghai 200433'.
2. On the 16th line of page 2, the Abstract 'Two-dimensional ~~(2D)~~ semiconductors ~~are booming and~~ have shown great potential for the development of advanced intelligent optoelectronic ~~integrated~~ systems. Among them, ~~2D two-dimensional~~ perovskite oxides with ~~wide bandgaps and~~ compelling optoelectronic performance have been ~~emerging as up to date frontiers~~ thriving in high-performance photodetection. However, harsh synthesis and defect chemistry severely limit their overall performance and ~~further~~ large-scale heterogeneous integration ~~in conventional top-down preparation processes~~. ~~In this work Here, by introducing a charge-assisted oriented assembly film formation (COAF) process, we report the~~ wafer-scale integration of highly oriented nanosheets ~~functional units is achieved by~~ introducing a charge-assisted oriented assembly film-formation process. ~~The and confirm its universality and scalability of this strategy for different types of 2D perovskite oxides are confirmed~~. The shallow-trap dominance induced by structural optimization endows the device with a distinguished performance balance, including photosensitivity close to that of single nanosheet units and the fastest response speed reported to date. An integrated ultra-flexible 256-pixel device demonstrates the versatility of material-to-substrate integration and ~~conformal imaging the functionality of the fabricated device~~. Moreover, the ~~performance enhancement enables the designed~~ device ~~to achieves~~ efficient recognition of multidirectional motion trajectories with an ~~high~~ accuracy of over 99.8%. Our work provides prescient insights into the large-area fabrication and utilization of 2D perovskite oxides in advanced optoelectronic ~~devices~~.'
3. On the 56th line of page 3, the sentence 'photoconductivity effect induced by the abundant traps leads to ~~extremely~~ slow response speeds'.

4. On the 103rd line of page 5, the sentence ‘the sprayed large number of micro-droplets first ensure the large-area in-plane spreading on the substrate-~~(Fig. 1a, i)~~. And the repulsive forces induced by the intrinsic negative charges between the nanosheets in micro-droplets prevent aggregation and facilitate their dispersion-~~(Fig. 1a, ii)~~. Then, when the droplet contacts the substrate, non-uniform repulsive forces between the nanosheets in the droplet and those on the substrate induce the former to rotate to be approximately parallel to the latter in order to balance the repulsive forces-~~(Fig. 1a, iii)~~, resulting in a highly ordered orientation-~~(Fig. 1a, iv)~~. As the solvent evaporates, the nanosheets in each micro-droplet settle and stack layer by layer, accompanied by the filling of voids and gaps between the larger nanosheets by the tinier ones (Supplementary Fig. 5). But given the high shape irregularity of the nanosheets, the formed films may still not be totally void-free. After the complete evaporation of solvent, tight structures with high degree of orientation are formed-~~(Fig. 1a, v)~~. Meanwhile, since the large number of micro-droplets cover the substrate surface uniformly over a large area during the spraying process, the nanosheets within each droplet will connect with those adjacent ones to form an integral whole. Ultimately, the ordered stack and the connections covering the whole substrate of the nanosheets simultaneously completes the construction of the highly oriented film-~~(Fig. 1a, vi)~~.’.
5. On the 128th line of page 6, the sentence ‘Optical microscope photographs and SEM characterization of surface morphology in Supplementary Figs. 6 and **Supplementary Fig. 7**’.
6. On the 143rd line of page 7, the sentence ‘With TEM and XPS tests, we demonstrate the purity and homo-disperse of the elements (Supplementary Figs. 9 and **Supplementary Fig. 10**)’.
7. On the 204th line of page 9, the sentence ‘With a channel area of 50,000 μm^2 ’.
8. On the 206th line of page 9, the sentence ‘remarkable detectivity of 11.9 A/W^{-1} and 3.71×10^{14} Jones’.
9. On the 209th line of page 10, the sentence ‘as far as we know of 185,560

displays’.

10. On the 217th line of page 10, the sentence ‘endows them with high optical gain and ~~outstanding~~ photosensitivity, it also introduces ultraslow photoresponse’.
11. On the 227th line of page 10, the sentence ‘response speed of $t_r = (42.4 \pm 14.8) \mu\text{s}$ and $t_d = (1.77 \pm 0.40) \text{ms}$ ’.
12. On the 233rd line of page 11, the sentence ‘we quantify the trap density (n_{trap}) by the space-charge-limited current (SCLC) method’.
13. On the 237th line of page 11, the sentence ‘and the n_{trap} of S-CNO devices that calculated from trap-filled limit voltages (V_{TFL})’.
14. On the 281st line of page 12, the sentence ‘Schematic illustrations and photographs of the fabrication process are shown in Supplementary Figs. 22 and ~~Supplementary Fig. 23~~’.
15. On the 289th line of page 13, the sentence ‘with a low variation coefficient (C_v) of 17.6%’.
16. On the 301st line of page 13, the sentence ‘remain highly uniform with a C_v of 17.3%’.
17. On the 321st line of page 14, the sentence ‘we propose a ~~novel~~ model for motion trajectory recognition’.
18. On the 398th line of page 18, the sentence ‘More details can be found in Supplementary Figs. 22 and ~~Supplementary Fig. 23~~’.
19. On the 452nd line of page 20, the sentence ‘Nevertheless, the optimizer may encounter a local minimum due to the ~~extremely~~ low sample quality and high learning rate.’.
20. On the 461st line of page 20, the sentence ‘The data that support the findings of this work are ~~provided in the main text and the Supplementary Information. More relevant data are~~ available from the corresponding authors upon ~~reasonable~~ request. ~~Source data are provided with this paper~~’.
21. On the 465th line of page 20, the sentence ‘The codes and dataset that used for designing and training the convolutional neural network are available from the corresponding authors upon ~~reasonable~~ request’.

22. On the 562nd line of page 24, the sentence ‘This work is supported by National Natural Science Foundation of China (No. 62374035 (X.S.F.), 92263106 (X.S.F.), 62204047 (Z.Q.L.), 52425308 (X.S.F.), and 12211530438 (X.S.F.))’.
23. On the 582nd line of page 26, the sentence ‘**a**, Schematic of the Charge-assisted oriented assembly film-formation (COAF) process of the S-CNO film ~~(i to vi)~~. Specific interpretations are presented in Supplementary Note I. **b**, Surface morphology SEM images of the S-CNO film during the COAF process. Scale bar: 1 μm . ~~(i) Sedimentation of the nanosheets in the initial stage. (ii) Accumulated sedimentation of the nanosheets. (iii) Formation of uniform nanosheets film.~~’.
24. On the 594th line of page 27, the sentence ‘**g-i**, GIWAXS patterns of D-CNO films with incident angles of 0.1° (**g**), 0.2° (**h**), 0.4° (**i**). Note that all the GIWAXS patterns use the same color bar as in Supplementary Fig. 14’.
25. On the 603rd line of page 28, the sentence ‘Statistics of the V_{TFL} and n_{trap} calculated from every three hole-only D-CNO and S-CNO devices’.
26. On the 604th line of page 28, the sentence ‘Schematic diagrams of carrier recombination and trapping kinetics in D-CNO (~~left~~) and S-CNO (~~right~~) films. The ~~top diagrams schematic illustrations~~ show the band structure containing trap and recombination centers, and the ~~bottom diagrams line graphs~~ show the practical photo-switching response.’.
27. On the 625th line of page 31, the sentence ‘Schematic illustration of spatiotemporal vision imaging acquisition and motion recognition realization. ~~After acquiring high quality single frames through device sensing, the spatiotemporal image is obtained through time weighting by a designed program, followed by CNN processing for motion recognition.~~’.
28. We have revised the section headings in the manuscript and the subtitles in SI file.
29. We have revised the order of the sections in both manuscript and SI file.
30. We have removed the unnecessary italics and bold font marks in the main paper and SI file.